# Genome-wide CRISPR screens identify the YAP/TEAD axis as a driver of persister cells in *EGFR* mutant lung cancer
Matthias Pfeifer[1,6,7], Jonathan S. Brammeld[2,7], Stacey Price[2], James Pilling[3], Deepa Bhavsar[1], Anca Farcas[1], Jessica Bateson[2], Anjana Sundarrajan[1], Ricardo J. Miragaia[1], Nin Guan[1], Stephanie Arnold[1], Laiba Tariq[1], Michael Grondine[1], Sarah Talbot[1], Maria Lisa Guerriero[3], Daniel J. O'Neill[3], Jamie Young[2], Carlos Company[1], Shanade Dunn[1], Hannah Thorpe[1], Matthew J. Martin[1], Kimberly Maratea[4], Daniel Barrell[1], Miika Ahdesmaki[1], Jerome T. Mettetal[1], Functional Genomics Centre[5]*, James Brownell[1] & Ultan McDermott[1] ✉

Most lung cancer patients with metastatic cancer eventually relapse with drug-resistant disease following treatment and *EGFR* mutant lung cancer is no exception. Genome-wide CRISPR screens, to either knock out or overexpress all protein-coding genes in cancer cell lines, revealed the landscape of pathways that cause resistance to the EGFR inhibitors osimertinib or gefitinib in *EGFR* mutant lung cancer. Among the most recurrent resistance genes were those that regulate the Hippo pathway. Following osimertinib treatment a subpopulation of cancer cells are able to survive and over time develop stable resistance. These 'persister' cells can exploit non-genetic (transcriptional) programs that enable cancer cells to survive drug treatment. Using genetic and pharmacologic tools we identified Hippo signalling as an important non-genetic mechanism of cell survival following osimertinib treatment. Further, we show that combinatorial targeting of the Hippo pathway and EGFR is highly effective in *EGFR* mutant lung cancer cells and patient-derived organoids, suggesting a new therapeutic strategy for *EGFR* mutant lung cancer patients.

Drug resistance is ultimately the cause of treatment failure for most cancer patients. This points to the existence of residual (or persistent) cancer cells, creating a reservoir that ultimately gives rise to stable drug resistance. These drug-tolerant persisters have been described for over a decade in numerous studies and may exploit non-genetic (transcriptional) programs that enable cancer cells to survive drug treatment although their biology is still poorly understood[1].

Approximately 10-20% of lung adenocarcinoma tumours harbour activating mutations in *EGFR*. Targeting these mutations clinically with EGFR tyrosine kinase inhibitors such as gefitinib or osimertinib improves survival, however, in the metastatic setting, almost all patients invariably develop drug resistance[2]. Previous studies have identified a number of

(mostly genetic) resistance mechanisms at disease progression including secondary mutations in *EGFR* itself (T790M, C797S), gene amplifications (*MET, FGF*), oncogenic mutations (*KRAS, PIK3CA*), gene fusions (*BRAF, RET*) or histologic transformations (small cell or squamous)[3,4]. However, the mechanisms underpinning cancer cell persistence shortly after treatment are less well understood, with recent studies highlighting a range of (non-genetic) processes ranging from epigenetic reprogramming, altered cell death thresholds, and more recently the YAP/TEAD axis[1,5]. Arguably, targeting such early persister cells may be a more effective strategy in preventing the evolution of drug resistance.

Forward genetic screens represent powerful tools to identify mechanisms of drug resistance – here we used genome-scale loss

[1]Oncology R&D, AstraZeneca, 1 Francis Crick Avenue, Cambridge CB2 0RE, UK. [2]Wellcome Sanger Institute, Hinxton, Cambridge CB10 1SA, UK. [3]Discovery Sciences, BioPharmaceuticals R&D, AstraZeneca, 1 Francis Crick Avenue, Cambridge CB2 0RE, UK. [4]Clinical Pharmacology & Safety, BioPharmaceuticals R&D, AstraZeneca, 1 Francis Crick Avenue, Cambridge CB2 0RE, UK. [5]AstraZeneca-Cancer Research Horizons Functional Genomics Centre, Jeffrey Cheah Biomedical Centre, University of Cambridge, Cambridge CB2 0AW, UK. [6]Present address: Leibniz-Institute of Virology (LIV) and University hospital Hamburg-Eppendorf (UKE), Hamburg, Germany. [7]These authors contributed equally: Matthias Pfeifer, Jonathan S. Brammeld. *A list of authors and their affiliations appears at the end of the paper. ✉e-mail: ultan.mcdermott@astrazeneca.com

(CRISPRn) and gain (CRISPRa) of function CRISPR screens to identify genes and pathways complicit in resistance (or persistence) to EGFR inhibitors in *EGFR* mutant lung cancer. This study represents one of the most comprehensive functional genomics studies of drug resistance in *EGFR* mutant lung cancer in both the 1st and 2nd line treatment settings. We show that resistance is mediated by a limited number of conserved pathways and that – most strikingly - a substantial number of resistance genes converge on the Hippo pathway. The Hippo pathway consists of a kinase cascade that regulates TEAD-dependent transcription by phosphorylation of the co-activators YAP1 and WWTR1 (TAZ) such that Hippo inactivation results in increased expression of YAP1/WWTR1/TEAD target genes. These genes regulate a diverse array of cellular programs, including proliferation, polarity, cell adhesion, and survival[6]. We show that YAP1/WWTR1/TEAD-dependent transcription is acutely activated following treatment of *EGFR* mutant lung cancer cells with EGFR inhibitors and that prevention of YAP1/WWTR1 activation strongly suppresses cancer cell persistence. In addition, both genetic and pharmacologic inhibition of YAP1/WWTR1/TEAD-dependent transcription re-sensitizes resistant cells to osimertinib as well as preventing the future emergence of persister cancer cells. Consequently, we propose Hippo signalling as a target mechanism for the prevention of osimertinib resistance and that inhibitors of the YAP1/WWTR1/TEAD axis should be explored as rational combination partners.

## Results

### Genome-wide CRISPRn and CRISPRa screens define a landscape of drug resistance in *EGFR* mutant lung cancer

Genome-wide CRISPR knockout (CRISPRn) and activation (CRISPRa) screens were performed in the *EGFR* mutant lung cancer cell lines PC-9 and HCC827 as well as isogenic clones (PC-9$^{T790M}$ and HCC827$^{T790M}$) harbouring the secondary *EGFR* T790M resistance mutation (which is responsible for resistance in the majority of patients treated with gefitinib in the 1st line setting). Each of the 4 cell lines was treated with IC90 concentrations of the EGFR inhibitors gefitinib and osimertinib which strongly inhibited signalling and cell viability (Fig. 1a, b; Supplementary Figs. 1a–h).

The MAGeCK algorithm was used to identify the most statistically significant and strongest resistance genes by comparison of treatment versus control arms across all 6 studies (*p* value <0.005 and absolute fold change of ≥2) (Fig. 1b, Supplementary Data 1 and Supplementary Data 2–7). Using these cut-offs, we identified a total of 1191 putative resistance genes in at least one of the 6 experiments. Each drug screened typically resulted in approx. 250–280 resistance hits per cell line. We reasoned that those genes detected as recurrent resistance hits in multiple screens ('core resistance genes') would be those more likely to be clinically relevant - 38 core resistance genes were detected in at least 4 of 6 experiments (Fig. 1c, Supplementary Fig. 2). These core resistance genes included a number of candidates previously clinically validated as resistance mechanisms including MET, KRAS, PTEN, and NTRK1[7]. We used the set of 38 core resistance genes to build a signaling network based on protein-protein interactions and pathway enrichments. Key pathways/processes significantly enriched for resistance genes included PI3K and MAPK, mTOR, ubiquitination, and Hippo signaling (Fig. 1d). Four of the top 10 most recurrent resistance genes (FOSL1, NF2, WWTR1 and PARD3) are modulators of YAP1/WWTR1-dependent transcription and Hippo signaling (Supplementary Data 1).

### Validation of drug-resistance genes

We selected a subset of the loss-of-function and gain-of-function resistance genes to validate experimentally in the cell lines PC-9 and HCC827 as well as in an additional *EGFR* mutant cell line, HCC4006 (Supplementary Figs. 1i, j). We overexpressed or silenced genes detected as either strong, moderate, or weak resistance drivers in the primary CRISPR screens (KCTD5, PTEN, NF1, MED24, CSK, and MET) (Supplementary Fig. 4a). In clonogenic survival assays using concentrations of osimertinib shown to efficiently inhibit signal transduction and cell viability, silencing or

overexpression of these genes confirmed the results of the CRISPR screens and also the strength of the resistance effect (Supplementary Figs. 4b–d).

### High-content microscopy combined with CRISPR determines pathway modulation by resistance genes

To better understand the mechanisms by which these genes were able to cause resistance we developed a plate-based CRISPR knockout platform that combines synthetic guides for gene silencing with high-content microscopy and quantification of immunofluorescent antibodies that represent the PI3K, MAPK, mTOR, or Hippo pathways. We measured pAKT (PI3K), pERK1/2 (MAPK), pS6 (mTOR), and nuclear YAP1/WWTR1 (Hippo) in PC-9, HCC827 and HCC4006 cells following acute 24 hr treatment with osimertinib and after cells had been transduced with synthetic guides targeting up to 72 recurrent resistance genes (Fig. 2a). We reasoned that a number of these genes might cause resistance by maintaining signaling through these pathway nodes following osimertinib treatment relative to non-transduced parental cells. Treatment of cells transduced with non-targeting control (NTC) synthetic guides with osimertinib showed an expected strong loss of signal from pERK1/2, pAKT, and pS6 (Fig. 2b–d). However, following targeting of the 72 resistance genes we observed that only a very limited subset targeted for silencing were capable of preserving signalling through the PI3K, MAPK, and mTOR pathways, all known mediators of resistance in *EGFR* mutant lung cancer (Fig. 2b–d, Supplementary Data 8–10).

Conversely, the most frequently observed outcome in all 3 cell lines tested following knockout of resistance genes was increased nuclear localisation of YAP1/WWTR1 – 13 of 65 genes (20%) in PC-9, 9 of 34 genes (26%) in HCC827 and 29 of 72 genes (40%) in HCC4006 were associated with increased nuclear localisation of YAP1/WWTR1 (defined as >20% increase relative to control cells) (Fig. 2b–d). Eleven resistance genes were associated with increased nuclear YAP1/WWTR1 in at least 2 of 3 cell lines, including known regulators of Hippo signalling (NF2, LATS2 and WWC1) as well as genes not previously known to be associated with the Hippo pathway (C16orf72, CAB39, MED12, DDA1, PPM1F, RNF7, SOX4). The majority of genes that increased nuclear YAP1/WWTR1 expression did not significantly alter pAKT or pERK signal intensity (Fisher's exact test), suggesting they mediate resistance through alternate processes. Interestingly, we also observed in parental PC-9 cells that 24 hours of osimertinib treatment was associated with increased nuclear localisation of YAP1/WWTR1 (with no change in total expression levels, Supplementary Fig. 4e); suggesting that modulation of Hippo signalling is an acute response to EGFR inhibition in *EGFR* mutant lung cancer and might cause drug resistance (Fig. 2e).

### Activation of YAP1/WWTR1/TEAD signalling causes osimertinib resistance

To confirm a role for Hippo signalling in osimertinib resistance, we silenced (KO) or overexpressed (OE) known positive and negative regulators of the pathway (NF2 KO, YAP1 OE, and WWTR1 OE) in *EGFR* mutant cell lines (Supplementary Fig. 5, Supplementary Fig. 6). NF2 is one of the best characterised negative regulators of Hippo transcriptional targets and was detected as a strong resistance hit in all 6 of the CRISPR knockout studies (Supplementary Data 1). We confirmed that knockout of NF2 in each of the cell lines PC-9, HCC827, and HCC4006 caused resistance to osimertinib and also increased expression of known canonical Hippo transcriptional targets (Fig. 3a). Silencing of NF2 was not associated with altered PI3K or MAPK signalling post osimertinib treatment as an explanation for the observed resistance, although increased basal phosphorylation of EGFR was observed as a result of NF2 knockout and may account for the increased proliferation rate of these cells (Fig. 3b)[8].

We used two systems to confirm that the NF2 loss in these cell lines was likely to be activating Hippo-dependent transcriptional programs, namely the detection of increased nuclear YAP1 and WWTR1 (a prerequisite for transcriptional activity) as well as a TEAD luciferase reporter system. This TEAD reporter is only active when YAP1 or WWTR1 is present in the

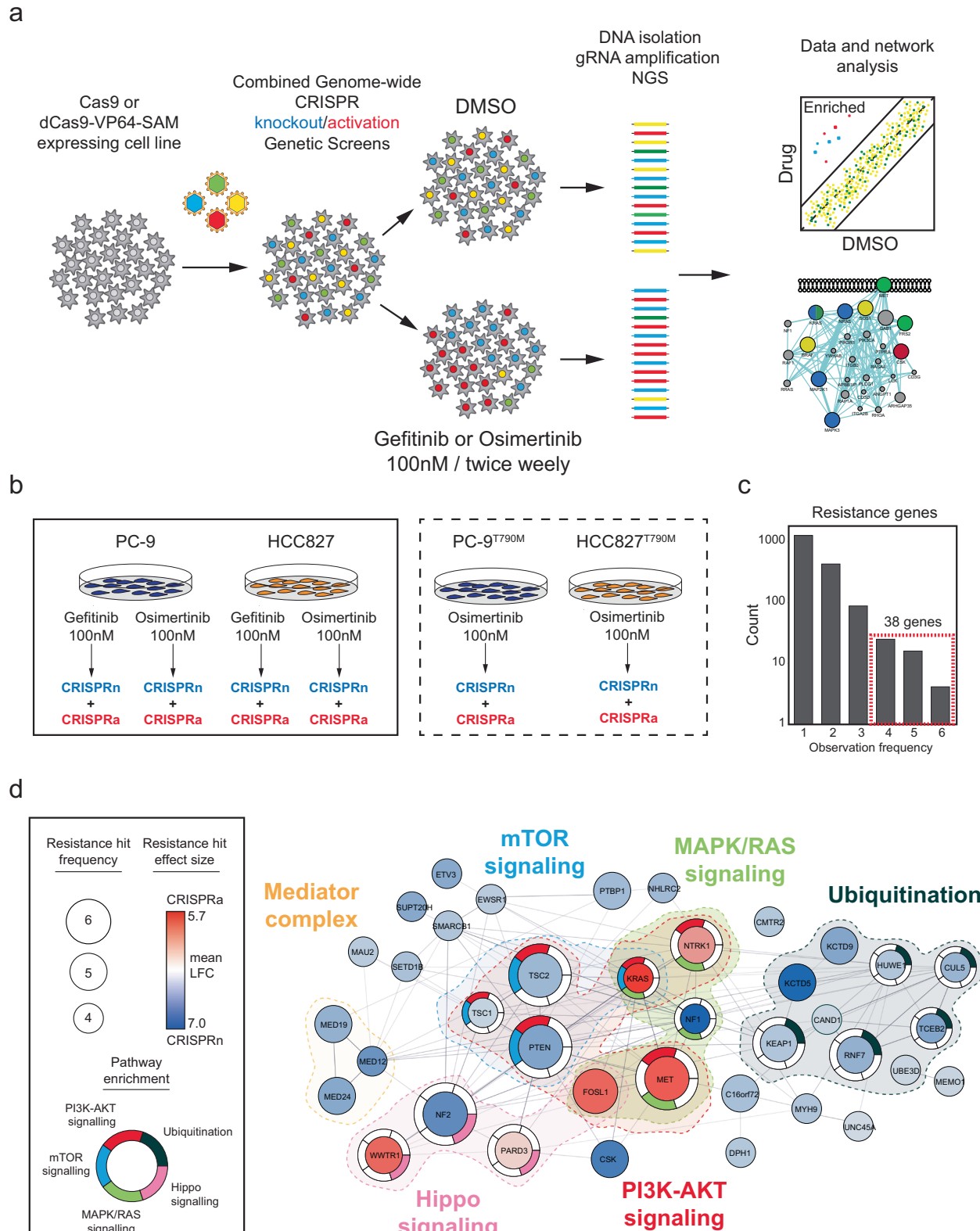

**Fig. 1 | Genome-wide CRISPRn and CRISPRa screens in *EGFR* mutant lung cancer. a, b** Scheme of experimental design, cell lines, and drugs used in this study. Transduced cells were cultured in gefitinib or osimertinib (IC80-90, 100 nM, twice per week) or vehicle control for 14 days, and gRNA abundance was measured using next-generation sequencing. Resistance genes were identified comparing enriched gRNAs in drug vs. vehicle-treated cells. **c** Bar graph of recurrence of resistance hits (Fold Change >2 and *p* value < 0.005). 38 genes were observed in at least 4 of the 6 experiments and defined as core drug resistance genes. **d** Key pathways enriched for resistance genes. Genes are coloured according to whether loss (blue) or gain (red) of function causes resistance.

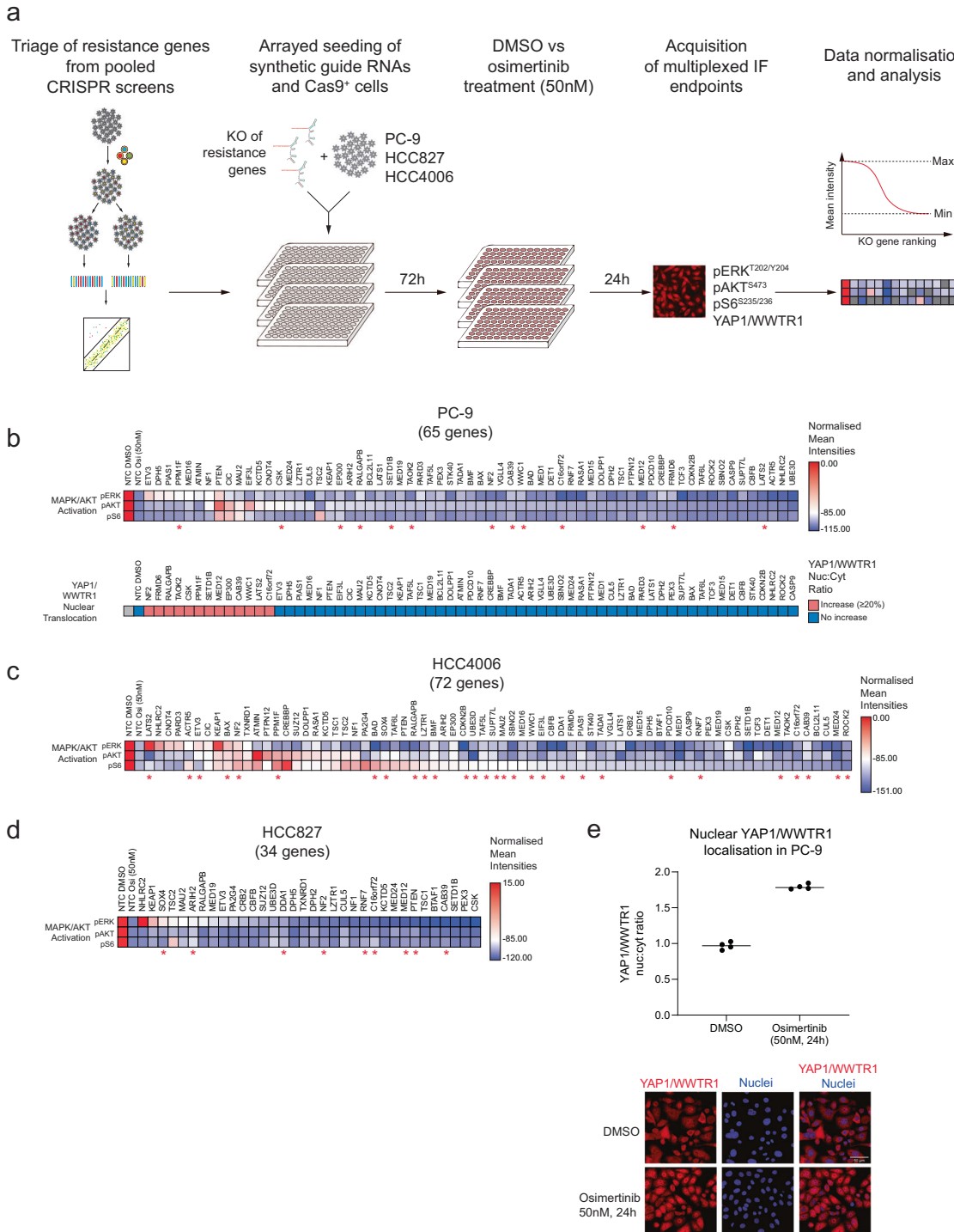

**Fig. 2 | High-content microscopy and CRISPR demonstrate that the Hippo pathway is an important component of the osimertinib resistance landscape.** **a** Assay setup for an arrayed CRISPR KO screen combined with high-content microscopy. Pools of synthetic guide RNAs targeting selected resistance hits from genome-wide screens were transfected in stably Cas9 expressing cell lines PC-9, HCC827, and HCC4006 and cultured for 72 h. Subsequently, cells were treated for 24 h with DMSO or osimertinib (50 nM). In a multiplexed approach, cells were imaged for expression of pathway markers pAKT (S473), pERK1/2 (T202/Y204), and pS6 (S235/236) as well as for nuclear localisation of the transcriptional co-activators YAP1/WWTR1 (Hippo pathway). Results were analysed using a max/min normalisation for pAKT, pERK1/2, and pS6, or the nuclear vs.

cytoplasmatic ratio of YAP1/WWTR1 expression. (**b-d**) The normalised effect of KO genes on pathway reactivation compared to DMSO (=0) or osimertinib-treated cells (= −100) (upper heatmap for each cell line). The first two columns in the upper heatmap for each cell line show the effect of osimertinib treatment alone on pERK, pAKT, and pS6 expression, to allow comparison with the effect of gene KO. In addition, red asterisks indicate genes that also show increased nuclear localisation of YAP1/WWTR1 after KO of resistance genes. All analyses are based on at least two replicates. (**e**) Increased nuclear vs. cytoplasmic YAP1/WWTR1 expression in PC-9 cells at 24 hours post osimertinib treatment. The method to calculate this is detailed in Methods.

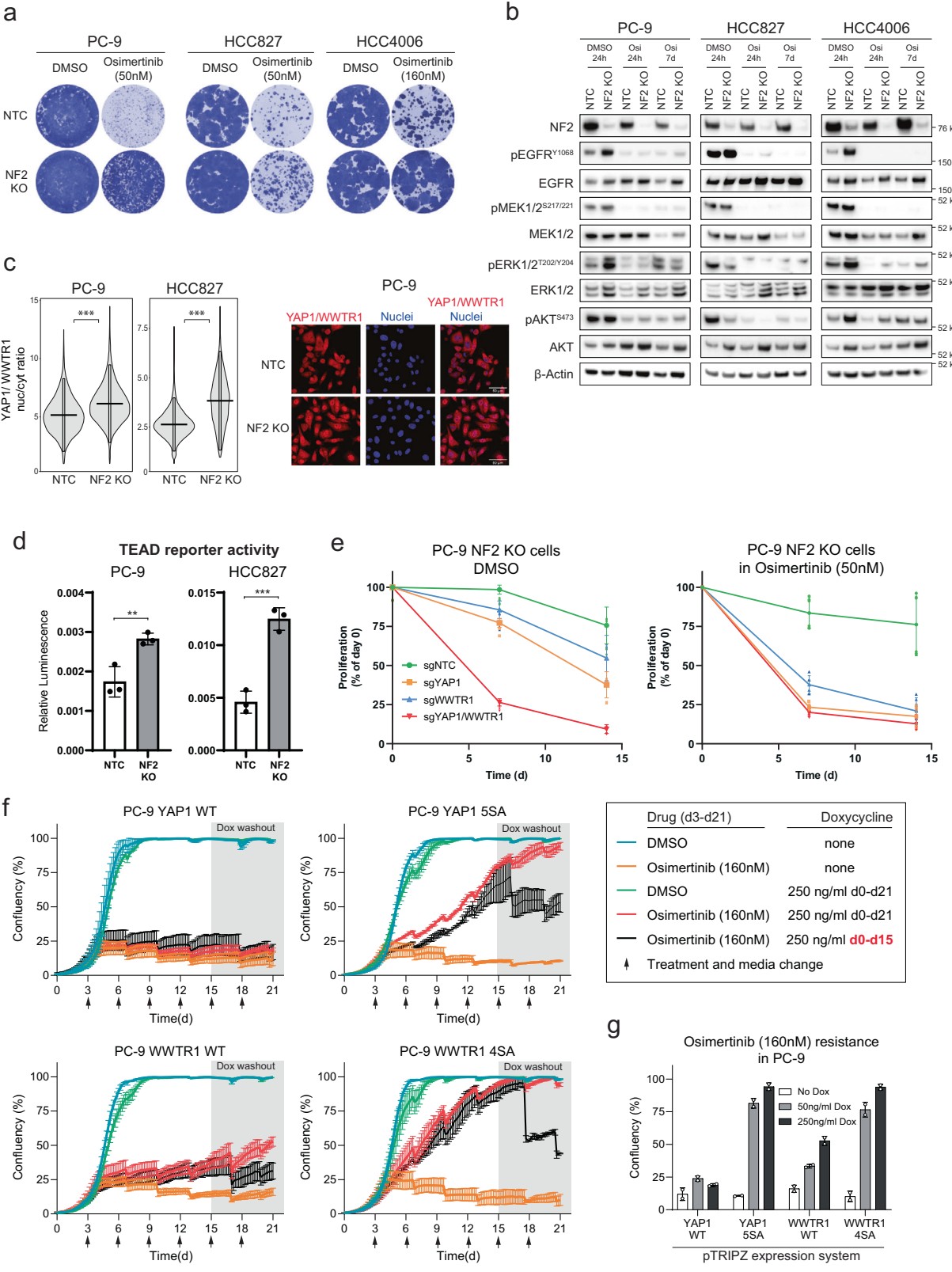

**Fig. 3 | The Hippo pathway modulates osimertinib response in *EGFR* mutant lung cancer.** Isogenic cell lines PC-9, HCC827, and HCC4006 were used to determine the effect of NF2 knockout on: **a** osimertinib resistance in 21-day clonogenic survival assays, **b** key EGFR signaling pathway post osimertinib treatment (24 hours) by western blot, **c, d** nuclear YAP1/WWTR1 expression (Wilcoxon rank sum test, ***$p < 0.001$) and activation of a TEAD reporter system (Student's *t*-test, **$p < 0.01$, ***$p < 0.001$). The right panel of (C) shows representative IF images of YAP1/WWTR1 fluorescence in NF2 KO vs NTC cells. **e** Effect on viability over 15 days in NF2 KO PC-9 cells when YAP1, WWTR1, or YAP1/WWTR1 were silenced and cells treated with DMSO vs osimertinib. **f** Effect in PC-9 cells of doxycycline-dependent expression of wild-type or mutant YAP1 or WWTR1 on viability following osimertinib treatment over 21 days. **g** Dose dependency effect of doxycycline on osimertinib resistance at 21 days in PC-9 cells for wild-type and mutant forms of YAP1 and WWTR1.

nucleus and bound to members of the TEAD family of transcription factors[9]. We confirmed that NF2 knockout was associated with significantly increased levels of nuclear YAP1/WWTR1 (Fig. 3c, Wilcoxon rank sum test). It also significantly increased TEAD reporter activity in PC-9 and HCC827 (Fig. 3d, Student's t-test). We observed that silencing either YAP1 or WWTR1 separately (as well as both together) profoundly re-sensitized the NF2 knockout cells to osimertinib (Fig. 3e, Supplementary Fig. 5b). We also observed that even in the absence of a drug, targeting the Hippo pathway in these cells (especially both genes together) also had a strong viability effect, indicating that loss of NF2 causes a dependency on the Hippo pathway (Fig. 3e, left panel).

YAP1 and WWTR1 are key activators of Hippo pathway transcriptional targets. We stably overexpressed full-length protein in PC-9 and HCC827 cells and confirmed resistance to osimertinib, although this effect was noticeably weaker in PC-9 cells for YAP1 (Supplementary Fig. 6a, b). Of note, YAP1 was not detected as a significant resistance gene from the CRISPR activation screens in PC-9 cells. We next used a doxycycline-inducible vector to over-express in PC-9 cells either wild-type YAP1 or WWTR1, as well as mutant versions where specific phosphorylation sites have been mutated to prevent inactivation of either gene (YAP1 5SA and WWTR1 4SA) (Supplementary Fig. 6c)[10,11]. Over-expression of wild-type WWTR1 caused increased resistance to osimertinib (red lines) which was reversible after a washout of doxycycline (black lines), compared to cells not treated with doxycycline (orange lines) (Fig. 3f, g). The effect was much weaker for wild-type YAP1. For wild-type WWTR1 we also observed a dose dependency effect – increased doxycycline levels (and therefore protein induction) was associated with increased resistance (Fig. 3f, Supplementary Fig. 6c). Both the YAP1 5SA and WWTR1 4SA mutants were shown to cause strong resistance to osimertinib (Fig. 3f, g).

### Components of the Hippo pathway are the most recurrent sensitizers to osimertinib in *EGFR* mutant lung cancer

Although the primary purpose of the CRISPR screens was to detect resistance genes, whether, through overexpression or silencing, the same data can be analysed for genes that when silenced have the opposite effect – enhancing the effect of osimertinib or gefitinib in *EGFR* mutant lung cancer cells. We constructed STRING networks from the top 20 most significant sensitizing genes to either drug in the cell lines and in 4 of the 6 cell lines screened, modifiers of Hippo signalling were among the most recurrent sensitizers (Supplementary Data 11).

### A diverse array of transcription factors mediate Hippo signaling but converge to drive an EMT program in *EGFR* mutant lung cancer

The TEAD transcription factor family (TEADs 1-4) is best known for mediating the transcriptional output of nuclear localisation of YAP1 and WWTR1 and ultimately Hippo signaling. Each TEAD has a specific tissue expression and plays different roles in development[12]. TEADs may also have different roles in mediating drug resistance and may be differentially active depending on whether YAP1 or WWTR1 are orchestrating the activation of downstream transcriptional programs. In addition, other transcription factors (TFs) may also be critical for executing the cell survival and proliferative effects of the Hippo pathway to cause drug resistance.

We used RNA-sequencing of isogenic models of the *EGFR* mutant cell lines PC-9, HCC827, and HCC4006 (NTC, NF2 KO, WWTR1 OE or YAP1 OE) to activate YAP1/WWTR1/TEAD-dependent transcription and identify all significant differentially expressed genes (Supplementary Data 12). These differentially expressed genes were used to identify (a) activated TFs (from a curated set of 222) and (b) altered downstream transcriptional programs/pathways.

We first confirmed by gene set enrichment analysis (GSEA) that each of these gene modifications (NF2 KO, YAP1 OE or WWTR1 OE) resulted in significant up-regulation of a canonical YAP1/WWTR1/TEAD-dependent transcriptional signature (Supplementary Fig. 7a, b)[10]. We then inferred TF activity based on the differentially expressed transcripts caused by NF2 KO,

WWTR1 OE, or YAP1 OE in each of the 3 cell lines (Fig. 4a; Supplementary Data 13). TEAD2 and TEAD4 were consistently identified in these isogenic models of Hippo signalling, with no evidence of TEAD1 activity (TEAD3 was not present in the curated set of TFs used). Components of the AP-1 complex (JUN, FOS), previously shown to associate with YAP1, WWTR1, and TEAD family members to activate TEAD-dependent transcription, were also detected although much less recurrently than TEADs. Furthermore, there was very little overlap between the TF activity resulting from knockout or activation of NF2, WWTR1 or YAP1 in the different cell lines (Fig. 4a, Venn diagrams), suggesting that cells can adopt a number of different routes, with significant redundancy, to ultimately activate canonical Hippo transcriptional programs (Fig. 4a). In PC-9 and HCC4006 cells, a relatively small set of TFs were the most strongly activated in our isogenic models, including TEAD2 and SP1 (both cell lines), JUN, NFE2, NR3C1 and PCBP1 (in PC-9) and MZF1, HIMFP, KLF4, KLF11 and TEAD4 (in HCC4006) (Fig. 4b–d). In HCC827, a much larger number of activated TFs were detected. Aside from TEADs, one of the strongest and most recurrently activated TFs observed across all 3 cell lines was SP1, a member of the Krüppel-like family of transcription factors. Of note, almost none of these TFs were identified as sensitizing to osimertinib (or gefitinib) when knocked out (as single genes) in our CRISPR screens (whereas YAP1 and WWTR1 were detected as significant sensitizing genes to both of these drugs). Interestingly, a similar analysis for TF activity in each parental cell line 48 h post osimertinib treatment identified TEAD4, SP1, and SP3 as the most recurrent and strongest activated TFs (Fig. 4e).

For each paired isogenic and parental cell line we calculated enrichment for Hallmark signature genes (Molecular Signatures Database v7.5.1) to identify altered transcriptional programs/pathways in response to knockout of NF2 or overexpression of YAP1 or WWTR1 (Supplementary Data 14a). The most recurrent significantly up-regulated pathway was that of Epithelial Mesenchymal Transition (EMT) (FDR < 0.2); the only hallmark significantly altered in every one of the isogenic models (as well as in response to osimertinib treatment in the parental cell lines) (Fig. 4f–h, Supplementary Fig. 6c). This highlights EMT as a potential common mechanism of osimertinib resistance triggered by activation of YAP1/WWTR1/TEAD-dependent transcription[5,13]. We also observed that this EMT transcriptional signature was enriched following osimertinib treatment in both PC-9 and HCC4006 parental cell lines and in all cell lines YAP1 and WWTR1 overexpression resulted in even greater enrichment for this state following treatment (Supplementary Data 14b).

Although YAP1 and WWTR1 are both synonymous with TEAD-dependent transcription, and indeed are often abbreviated as 'YAP1/WWTR1' in much of the literature, there is an increasing body of evidence that they can have very different functional effects depending on the experimental conditions[10]. We observed in our own CRISPR study that WWTR1 was detected much more frequently as a resistance gene than YAP1 in the *EGFR* mutant cell lines screened (Supplementary Data 1). In support of this difference, we observed that the differentially expressed genes in the YAP1 and WWTR1 overexpression models that contributed to the EMT expression signature, while overlapping to some degree for both genes, was often driven by genes uniquely expressed in either the YAP1 or WWTR1 models (Fig. 4f–h).

### The Hippo pathway drives resistance in persister cells in *EGFR* mutant lung cancer

Although we demonstrated that activation of YAP1/WWTR1/TEAD-dependent transcription is a potent driver of resistance to osimertinib in *EGFR* mutant lung cancer, there is no evidence that genes involved in the regulation of this pathway (such as NF2, LATS1/2, WWTR1, and YAP1) are commonly found as acquired resistance genes in patients who develop drug resistance. We reasoned therefore that if the Hippo pathway was important in mediating drug resistance in *EGFR* mutant lung cancer, this might be (a) through non-genetic mechanisms and (b) be involved at a much earlier stage of treatment response, i.e. acutely in maintaining the survival of drug-tolerant persister cells. Of note, we used whole exome and deep targeted

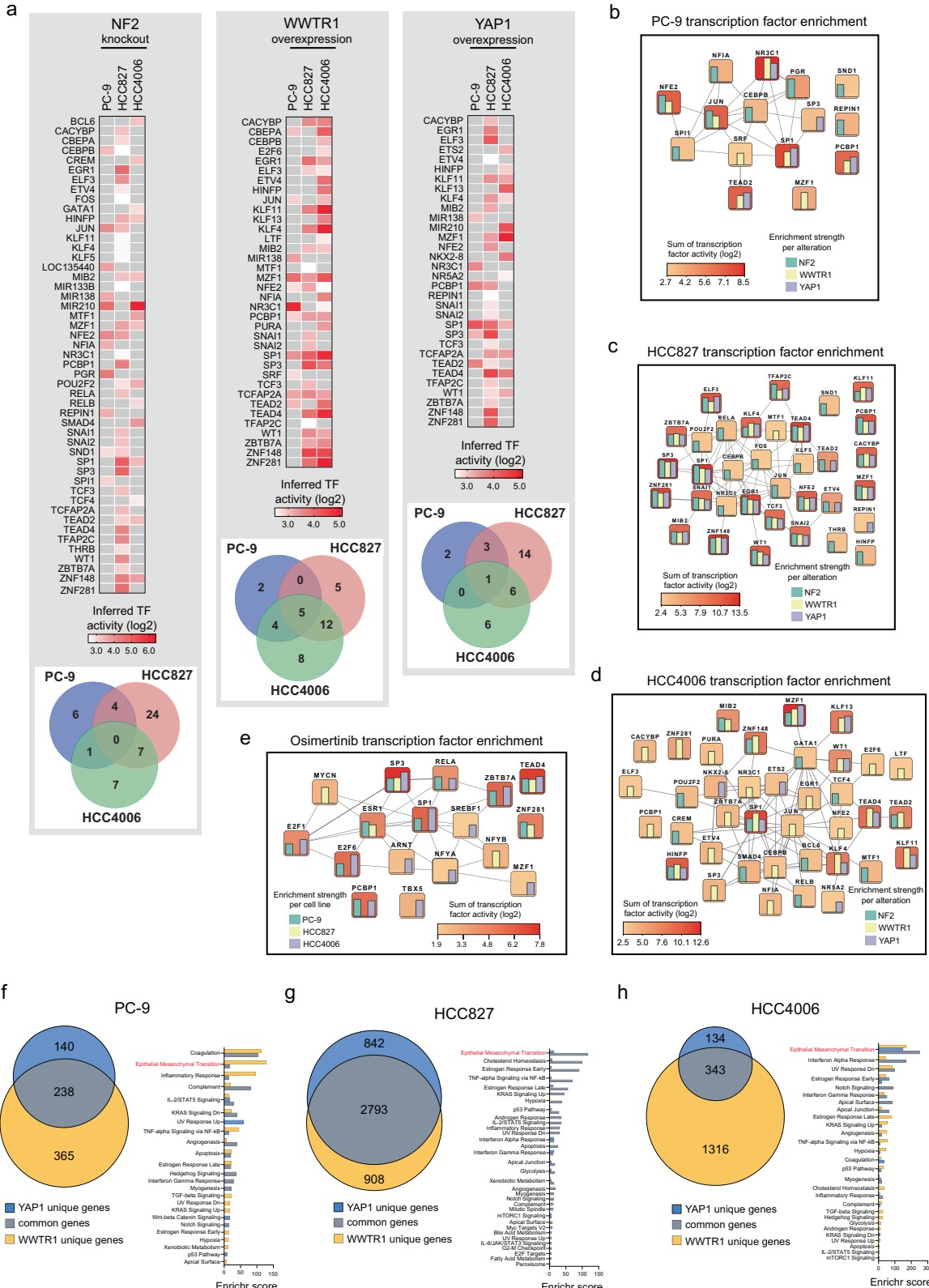

**Fig. 4 | The gene regulatory network of cancer cells following activation of the Hippo pathway. a** Transcription factor (TF) activity following activation of Hippo signaling through NF2 loss or overexpression of WWTR1 or YAP1 in *EGFR* mutant lung cancer cells. Cut-off for TF selection was $p < 0.01$ after enrichment using Enrichr. Heatmaps show inferred TF activity (log2 normalised). Scales are normalised to maximum and minimum of each heatmaps. **b–e** STRING interaction networks of activated TFs following (i) NF2 loss, (ii) overexpression of WWTR1 or YAP1, or (iii) osimertinib treatment in PC-9, HCC827 and HCC4006 cancer cells

(160 nM for 48 h). The mean activity score for each TF is indicated by the colour intensity of each box and the bars within each box represent individual elements that comprise that mean score. Number of deregulated genes (cut-off $p < 0.01$, FC $+$ / $-1.5$) after Hippo effector upregulation cause and enrichment of Hallmark pathways in (**f**) PC-9, (**g**) HCC827, and (**h**) HCC4006 cells. Cut-off for pathway selection was $p < 0.01$ after enrichment. Graphs show Enrichr score as a combined measure of significant enrichment and signature size.

gene panel sequencing to confirm that no pre-existing clonal or subclonal genetic mechanisms of resistance were present pre-treatment in our models.

We previously observed significantly increased nuclear localisation of YAP1 and WWTR1 as an acute response to osimertinib treatment (Fig. 2e). We therefore measured nuclear YAP1 and WWTR1 as well as expression of canonical Hippo pathway transcriptional targets following treatment with osimertinib in 3D organoid and 2D cell line models (Supplementary Data 16). Cells were treated with osimertinib at either a clinically relevant concentration (160 nM) or higher doses (500-1000 nM; to enrich better for drug-tolerant persister cells). In all *EGFR* mutant cell lines tested, there was a time-dependent and significant ($P < 0.001$, Wilcoxon rank sum test) increase in nuclear localisation of YAP1 and WWTR1 (Fig. 5a). We also detected increased expression by Q-PCR of the canonical Hippo pathway mRNA transcripts CTGF, CYR61, and AMOTL2 in 4 of the 5 models tested (3 cell lines and 2 organoids - cell lines shown here in Fig. 5b) (Supplementary Data 16). Unlike YAP1, which is primarily regulated by nuclear translocation, WWTR1 is primarily regulated by protein degradation and we confirmed an increase in total protein expression (and not mRNA) following osimertinib treatment in PC-9 cells (Supplementary Fig. 7c).

Finally, we tested if the EMT signature we observed as enriched following NF2 knockout or overexpression of YAP1 or WWR1 could also be detected following acute treatment of the parental *EGFR* mutant cells with osimertinib. We observed a strong enrichment of this EMT signature in all three cell lines (although not significant in HCC827) (Supplementary Fig. 7d; Supplementary Data 14).

Single-cell sequencing has also been used to detect altered transcriptional programs in persister cells and we analyzed data from two publicly available scRNA-seq time course datasets – where PC9 cells were treated respectively with osimertinib or erlotinib[14,15]. In both studies there was a significant enrichment in single cells post-treatment with the canonical Hippo activation signature (Day 0 vs any other day; Wilcoxon rank sum test, *p* value < 0.001) (Supplementary Fig. 8)[10].

### Combinatorial targeting of the Hippo pathway and EGFR is highly effective in *EGFR* mutant lung cancer cells

K-975 is a TEAD inhibitor shown to disrupt activation of Hippo transcriptional targets through the inhibition of protein-protein interactions between YAP1/WWTR1 and TEADs[16]. We confirmed using the TEAD luciferase reporter assay in cell lines engineered to activate YAP1/WWTR1/TEAD-dependent transcriptional programs (NF2 KO PC-9 and HCC827 cells) that K-975 disrupts this binding of YAP1 or WWTR1 to TEAD family members (Supplementary Fig. 7e). There was little evidence of a viability effect when used as a single agent in *EGFR* mutant lung cancer at concentrations shown to disrupt TEAD binding (Supplementary Fig. 7f).

In PC-9, HCC827, and HCC4006 cell lines, typically 5-10% of cells persist following osimertinib treatment (despite harbouring the target *EGFR* mutation). In a short-term assay, combining osimertinib with the K-975 TEAD inhibitor (at a single dose shown to disrupt YAP1/WWTR1 and TEAD binding) resulted in a dose-dependent and significant decrease in persister cancer cells (Fig. 5c). In a longer duration assay (21 days), combining K-975 with osimertinib at a clinically relevant concentration (160 nM) had an even more profound effect, almost completely ablating persister cancer cells compared to osimertinib alone, despite minimal activity from the TEAD inhibitor as monotherapy (Fig. 5d). This effect was also observed when we used a much higher concentration of osimertinib (500 or 1000 nM) to specifically enrich for drug-tolerant persister cells – the osimertinib/K-975 combination (red line) profoundly suppressed cancer cell viability compared to either single agent in all three of the cell lines tested and ablated any residual cells (Supplementary Fig. 7g).

We previously showed that activation of downstream Hippo pathway transcriptional programs through deletion of NF2 was associated with resistance to osimertinib in all *EGFR* mutant cell lines tested (Fig. 3a). Here, combining either K-975 (or an alternate TEAD inhibitor, MYF-01-037) with osimertinib resulted in either complete or a significant reversal of

osimertinib resistance in NF2 deleted PC-9, HCC827 or HCC4006 cancer cells (Supplementary Figs. 9a–g)[5].

### Increased nuclear YAP1 in a subset of patient-derived models following osimertinib treatment

A panel of 10 *EGFR* mutant lung cancer patient-derived xenografts (PDX) was used to characterize Hippo signalling in the setting of osimertinib treatment. The models were classified as sensitive or resistant based on their in vitro response to osimertinib (Fig. 6a; Supplementary Data 15). Of note, all 5 of the resistant models were acquired from patients progressing on osimertinib treatment and had acquired genomic alterations likely to cause resistance (Supplementary Data 15). We quantified nuclear YAP1 and WWTR1 IHC expression in the panel of *EGFR* mutant lung cancer PDX. Of note, each of the resistant models had a plausible acquired genomic alteration to explain resistance (Supplementary Data 15). We detected an increase in nuclear YAP1 expression in the osimertinib-resistant patient samples, although not statistically significant (Fig. 6b). There was no consistent pattern of expression of nuclear WWTR1 in the sensitive versus resistant samples.

To address whether in vivo there is evidence of increased expression of nuclear YAP1 or WWTR1 in patient-derived samples after osimertinib treatment, we engrafted the two osimertinib sensitive PDX models (CTG-2548 and LU5221) in immunodeficient NSG mice and treated daily with clinically relevant doses of osimertinib versus vehicle for 28 days. We confirmed in both models tumour regression during the 28 days of daily treatment but importantly when treatment was stopped, the tumours regrew – indicating the presence of viable persistent cancer cells despite treatment (Fig. 6c). Replicate samples collected pre-treatment and at the end of the 28 days of osimertinib treatment were used to quantify nuclear YAP1 and WWTR1 expression by IHC (Fig. 6c, Supplementary Fig. 10). In the CTG-2548 model, there was a significant increase in YAP1 nuclear expression ($p = 0.005$) (and a commensurate decrease in WWTR1 expression) following osimertinib treatment[17]. There was no evidence of induction of either YAP1 or WWTR1 nuclear expression in the LU5221 PDX model.

## Discussion

Drug resistance is a major challenge in the treatment of most cancers. With few exceptions, it is ultimately the cause of treatment failure in almost all patients diagnosed with metastatic disease; understanding the causes of resistance (and therefore how to avoid or reverse it) is a major activity for many research groups[18]. Although in the past much of that activity has been focused on the detection of genomic alterations in samples from patients after drug resistance has developed, the landscape of resistance drivers that have been detected point to a high degree of heterogeneity - different resistance mutations are possible in different metastases within the same patient[19]. Defining rational drug combinations in this late-stage setting to overcome resistance is therefore extremely challenging and has prompted many groups to begin to analyse cancer cells at a much earlier stage of the transition to drug resistance, namely in the setting of those cells that are persistent (or residual) following initial treatment. These 'drug tolerant persister cells' can exploit non-genetic (transcriptional) programs to avoid cell death and ultimately can develop into the stable genetic resistance we observe in patients at late relapse. A variety of different mechanisms that cause cancer cell persistence have been proposed, ranging from effects on cell cycle, tumour microenvironment interactions, adaptions of cell metabolism and changes in cell state[1]. Although a number of potential therapeutic approaches to target cancer persistence have been proposed, as yet none have been successfully translated into the clinic.

Genome-wide CRISPR/Cas9 screens are a powerful tool to identify those genes and pathways complicit in driving drug resistance. We reasoned that the landscape captured in this way for resistance pathways/complexes in *EGFR* mutant lung cancer might be relevant as cancer persistence mechanisms. We found that genes in the Hippo pathway were the most recurrent and strongest resistance drivers to osimertinib in *EGFR* mutant lung cancer. As none of these genes have been identified as common *genetic* resistance mechanisms to osimertinib in clinical studies of relapsed patients,

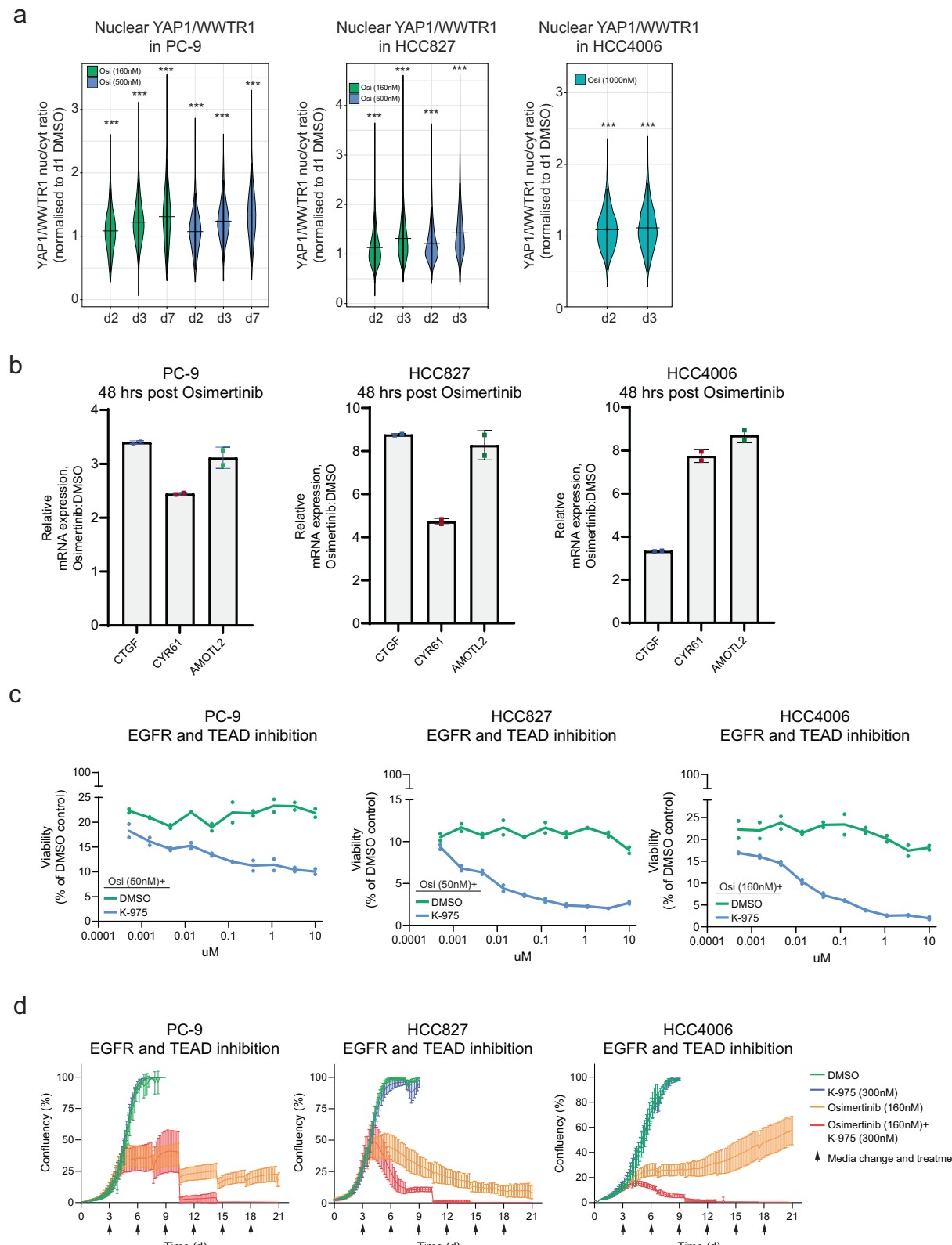

**Fig. 5 | The Hippo pathway drives resistance in persister cells in *EGFR* mutant lung cancer. a** Nuclear/cytoplasmic YAP1/WWTR1 expression ratio by immunofluorescence (normalised to day 1 DMSO) in PC-9, HCC927, and HCC4006 cells following osimertinib treatment over a time course (2, 3, and 7 days; Wilcoxon rank sum test, ***$P < 0.001$). **b** mRNA expression of Hippo transcripts CTFG, CYR61,

and AMOTL2 at 48 hours post osimertinib treatment. **c** 5-day viability assay of a concentration range of either K-975 or MYF-01-037 TEAD inhibitors combined with a fixed dose of osimertinib. **d** 21-day viability assay of osimertinib (160 nM) and K-975 (300 nM) alone versus in combination in PC-9, HCC827, and HCC4006 cells.

**Fig. 6 | Hippo signalling is activated acutely in patient-derived models following osimertinib treatment. a** Schema of patient-derived xenograft models, osimertinib response status, and assays used. **b** Nuclear YAP1 expression in a panel of 10 *EGFR* mutant lung cancer PDX models derived from patients classified as either osimertinib sensitive or resistant. Each dot indicates expression from a separate implanted mouse. The putative resistance mutation/copy number alteration is indicated above each resistant model. **c** (Left panel) Tumour volume measurements in *EGFR* mutant lung cancer PDX models CTG-2548 and LU5221 following 28 days of treatment with osimertinib in vivo; (Right panel) nuclear immunofluorescence of YAP1 and WWTR1 at timepoints t0 (pre-treatment) and t1 (28 days of treatment).

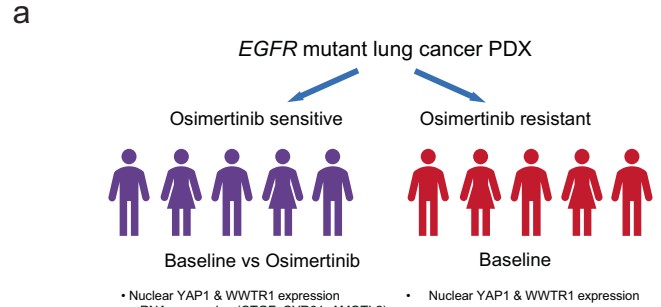

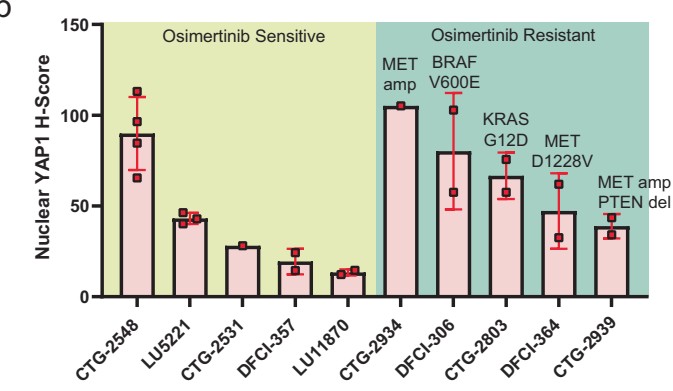

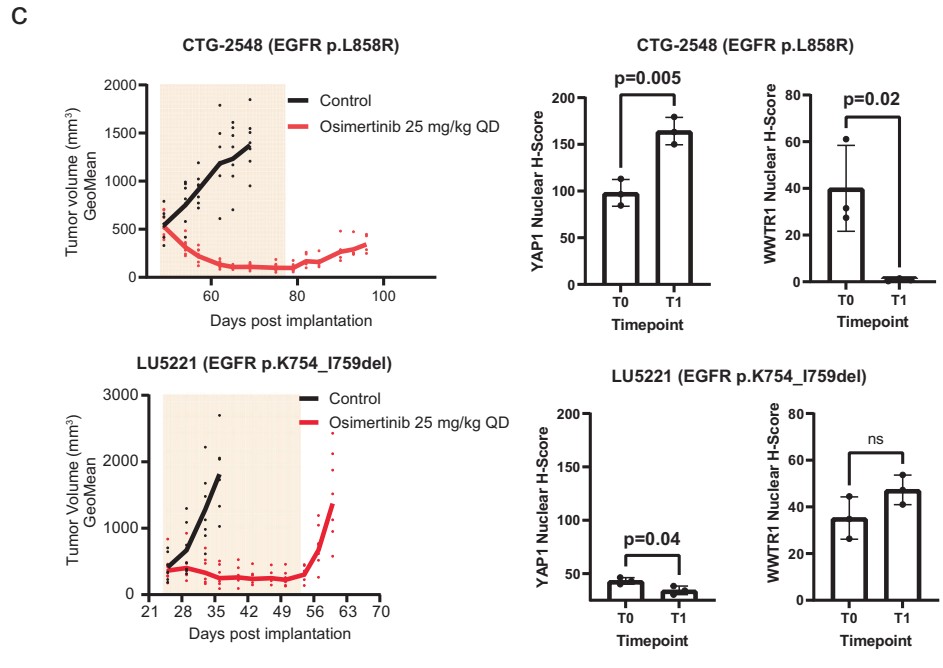

we investigated whether the Hippo pathway might instead be responsible for cancer cell persistence acutely after drug treatment. We confirmed both in vitro and in vivo that YAP1/WWTR1/TEAD-dependent transcription is acutely activated following osimertinib treatment in *EGFR* mutant lung cancer and pharmacologic and genetic ablation of this complex strongly suppresses persister cells. Of note, in the two PDX models we tested only one demonstrated YAP1/WWTR1/TEAD-dependent transcription in persister cells, highlighting that other mechanisms of persistence in *EGFR* mutant lung cancer must exist and need to be defined. EMT was the most frequently activated transcriptional program following osimertinib treatment. Acquisition of an EMT state in cancer cells has been shown to prevent cell death

and has been detected in patients who develop acquired resistance to EGFR inhibitors[20,21]. At present, it is not possible to effectively reverse the EMT cell state pharmacologically – an alternate approach to prevent an EMT transition would be through inhibition of YAP1/WWTR1/TEAD-dependent transcription. A number of selective inhibitors of the TEAD family members are in development and have been shown to disrupt YAP1/WWTR1/TEAD-dependent transcription, affect the viability of *NF2* mutant mesothelioma models, and to enhance the efficacy of EGFR inhibitors in *EGFR* mutant lung cancer[5,16,22].

There are limitations of this study – we found evidence that one of the most recurrent and strongest resistance pathways identified in these screens

(Hippo) is acutely activated following osimertinib treatment and targeting this pathway strongly reduces the survival of persister cells. Although we validated this finding in cancer cell lines using an osimertinib/TEAD inhibitor combination, lack of access to an inhibitor with in vivo efficacy at the time of performing these experiments precluded assessing whether we observed the same effect in vivo. We note that a recent AACR abstract does confirm that an osimertinib/TEAD inhibitor combination does suppress viability in vivo in *EGFR* mutant PDX and cell line xenograft models[22]. Further studies exploring the in vivo role of an osimertinib/TEAD inhibitor combination are warranted given the development of active inhibitors in animal models although arguably the most critical next experiments should be the assessment of the relevance of the Hippo pathway in patient samples using biopsies obtained post-treatment with osimertinib – whether neoadjuvant studies or in the metastatic setting. These are outside of the scope of this study but would confirm the clinical relevance of our findings. In addition, it was not possible in this study to assess the impact of Hippo pathway activation on elements of the tumour environment in the cell line or 3D organoid models tested which would require the clinical sample testing described above or the use of syngeneic or genetically-engineered mouse models.

Although we show activation of YAP1/WWTR1/TEAD-dependent transcription following osimertinib treatment in both in vitro and in vivo models, confirmation that this is observed in persister cells in patients clinically is still required – at present single-cell sequencing is the 'go to' technology for characterising rare persister cells with its ability to characterise the transcriptome at single cell resolution[15]. Importantly, patient samples would include components of the tumour microenvironment such as immune cells, fibroblasts, and macrophages that may be remodelled following drug treatment and support the persister cell population[23,24]. However, in practice there are significant logistic and technical hurdles: (1) biopsy of patients at maximum treatment response (typically 6-8 weeks in the case of osimertinib) may be difficult if the tumour has shrunk dramatically to make even a radiologically-guided biopsy difficult, (2) the biopsy sample may contain very few cancer (persister) cells with a large amount of stroma, (3) the persister cells may be extremely sensitive to any manipulation after biopsy prior to single-cell sequencing (dissociation of tissue to single cells, flow sorting, washing steps) and it can be challenging to recover viable cancer cells after such processing (although a newer protocol from 10X Genomics that paraformaldehyde fixes cells may resolve some of these issues).

Notwithstanding these challenges, a number of consortia globally have set out to create single-cell atlases that characterise how cancer cells transition over time (premalignant to malignant, primary to metastases, and untreated cells to persister cells) (https://humantumoratlas.org/; https://persist-seq.org/). Based on our data, we would propose that CRISPR screens may be useful in the interpretation of such transcriptomic data to identify those pathways/complexes that are potential novel vulnerabilities for persister cells.

## Methods
### Materials
All cell culture was performed in either RPMI or DMEM/F12 medium (according to the supplier's recommendations) and supplemented with 5% FBS and penicillin/streptavidin. Cells were maintained at 37 °C and 5% $CO_2$ during culture. Mycoplasma testing was carried out using the MycoSEQ Mycoplasma detection kit (ThermoFisher). PC-9 and HCC827 cell lines were purchased from ECACC and ATCC respectively. All cell line used in these studies were authenticated using STR profiling (IDEXX Laboratories). Clones harbouring the *EGFR* T790M resistance mutation were generated from the parental cell lines to allow us to model resistance following 1$^{st}$ line treatment with gefitinib. For PC-9, resistant clones were generated following serial weekly treatment with an IC90 concentration of gefitinib over 6 weeks; a number of groups having previously demonstrated the emergence of *EGFR* T790M mutant clones in PC-9 cells using this approach[25]. Individual clones were isolated using cloning cylinders, expanded and whole exome

sequenced. All clones harboured the EGFR T790M resistance mutation and one of the clones (PC9-GR5) was used in these studies and renamed for clarity PC-9$^{T790M}$. In the case of HCC827, it has been shown that a similar approach generated MET amplification as a resistance mechanism[26]. We therefore mutagenized HCC827 cells with the chemical mutagen ENU to generate point mutations and treated them weekly with an IC90 concentration of gefitinib over 6 weeks to generate stably resistant clones. Individual clones were isolated using cloning cylinders, expanded and whole exome sequenced to detect those harbouring the *EGFR* T790M mutation. One of these was selected for these studies and called HCC827$^{T790M}$. There was no evidence from whole exome sequencing that these clones had acquired any other driver mutations compared to the parental PC-9 and HCC827 cell lines. The PC-9 and HCC827 T790M mutant clones were confirmed to be resistant to gefitinib and sensitive to osimertinib (Supplementary Figs. 1A–D).

### Immunoblotting
Cells were collected at indicated time points and lysed using RIPA lysis and extraction buffer (ThermoFisher Scientific, #89901) supplemented with protease and phosphatase inhibitor cocktail (ThermoFisher Scientific, # 78441). Per sample 10ug lysate was separated via SDS-PAGE and transferred onto nitrocellulose membranes. Following primary and secondary antibodies were used for detecting protein abundance: anti-β-actin (Sigma, A5316), anti-CSK (Cell Signaling Technology, #4980), anti-KCTD5 (Proteintech, 15553-1-AP), anti-NF1 (abcam, ab17963), anti-NF2 (Cell Signaling Technology, #6995), anti-PTEN (Cell Signaling Technology, #9559), anti-YAP1 (Cell Signaling Technology, #14074), anti-WWTR1 (Cell Signaling Technology, #70148), anti-p-EGFR$^{Y1068}$ (Cell Signaling Technology, #2234), anti-EGFR (Cell Signaling Technology, #4267), anti- p-AKT$^{S473}$ (Cell Signaling Technology, #4060), anti-AKT (Cell Signaling Technology, #9272), anti-p-MEK1/2$^{S217/221}$ (Cell Signaling Technology, #9154), anti-MEK1/2 (Cell Signaling Technology, #4694), anti-p-ERK1/2$^{T202/Y204}$ (Cell Signaling Technology, #9106), anti-ERK1/2 (Cell Signaling Technology, #9102), anti-MET (Cell Signaling Technology, #8198), anti-GAPDH (Cell Signaling Technology, #2933), anti-mouse-IgG-HRP (Cell Signaling Technology, #7076) and anti-rabbit-IgG-HRP (Cell Signaling Technology, #7074). Immunoblot raw data has been provided (Supplementary Fig. 12).

### Genome-wide CRISPR/Cas9 screens
For genome-wide loss-of-function CRISPRn screens, cell lines were transduced with a sgRNA library targeting 18,010 human genes[27]. For genome-wide gain-of-function (activation) CRISPRa screens, cell lines were transduced with a sgRNA library targeting 23,430 coding isoforms with a unique transcription start site[28]. After selection, library-transduced cells were treated with either gefitinib or osimertinib (IC80-90, 100 nM, twice per week) over 14 days to select resistant cells.

**Lentiviral genome-wide gRNA library construction.** pKLV2-U6gRNA5(BbsI)-PGKpuro2ABFP-W was used (Addgene, #50946). Packaging plasmids, psPax2 and pMD2.G (Addgene #12260, #12259) were used at the following mixing ratio: 5.4 µg lentiviral vector, 5.4 µg psPax2 and 1.2 µg pMG2.G per 10 cm dish. Transduction of all cells was performed in 6-well plates as follows: $1 \times 10^6$ cells and viral supernatant were mixed in 2 ml of culture medium supplemented with 8 µg/ml (human) Polybrene (Millipore), and incubated overnight at 37 °C. The medium was refreshed on the following day and the transduced cells were cultured further.

**Generation of Cas9-expressing cancer cell lines.** Cell lines were transduced with lentivirus produced from the Cas9 pKLV2-EF1a-Cas9Bsd-W expression vector (Addgene #68343). Blasticidin selection was initiated 2 days after transduction at 50 µg/ml. To assess the ability of Cas9-expressing cells to efficiently silence full-length gene expression, cells were transduced with a lentivirus produced from the Cas9 reporter vector ('Cas9 reporter vector') – the pKLV2-U6gRNA5(gGFP)-

PGKBFP2AGFP-W vector (Addgene #67980). This vector contains both a GFP expressing cassette as well as a gRNA targeting GFP – efficient Cas9 activity would therefore be expected to result in silencing of GFP signal. The ratio of BFP only and GFP-BFP-double positive cells were analysed on a BD LSRFortessa instrument (BD) 3 days post-transduction for cancer cells. The data were subsequently analysed using FlowJo.

**Generation of genome-wide CRISPR/Cas9 libraries and positive selection screens.** $6 \times 10^7$ cells were infected with a pre-determined volume of the genome-wide gRNA lentiviral supernatant to ensure transduction MOI of 0.3, at which most cells receive only one genetic perturbation and therefore the gRNA library has a coverage of >200 cells expressing each gRNA. Each cell line was transduced as duplicates. Two days after transduction, the cells were selected with puromycin at 2-3 µg/ml for 5-6 days and further cultured. Following selection, cells were maintained in culture for 14 days to allow for complete depletion of protein products of targeted genes, following which cells were treated with either DMSO (control), gefitinib, or osimertinib (100 nM, drug replaced twice per week). After 14 days of drug selection, cells from each of the duplicate DMSO and treatment arms were harvested and submitted separately for PCR and Illumina sequencing. The DMSO cells were used as a control for genes that when silenced increase the proliferation rate of cells and therefore would contribute disproportionately to gRNA enrichment in the drug-treated cells.

**Illumina sequencing of gRNAs and statistical analysis.** Genomic DNA extraction and Illumina sequencing of gRNAs were conducted[29]. In brief, 72ug of total extracted DNA was used to set up 36 PCR reactions (2ug each) using 10uM concentrations of forward and reverse primers following which PCR products were purified using spin columns before a second PCR reaction was carried out to incorporate indexing primers for each sample. DNA was purified using SPRI beads and submitted for Illumina sequencing.

F primer: ACACTCTTTCCCTACACGACGCTCTTCCGATCTCT TGTGGAAAGGACGAAACA.

R primer: TCGGCATTCCTGCTGAACCGCTCTTCCGATCTC-TAAAGCGCATGCTCCAGA.

Enrichment and depletion of guides and genes were analyzed using the MAGeCK RRA (version 0.5.8) statistical package by comparing read counts from each cell line with counts from matching DMSO cells, after comparing each to counts from the plasmid gRNA library (https://sourceforge.net/projects/mageck/)[30]. MAGeCK RRA ranks sgRNAs based on their *p* values calculated from the negative-binomial model and uses a modified RRA algorithm named α-RRA to identify positively or negatively selected genes.

**Genome-wide CRISPR Activation screens**
**Generation of MS2-p65-HSF1 and dCas9-VP64 stably expressing cancer cells.** The human CRISPR SAM library consists of three components: a nucleolytically inactive Cas9-VP64 fusion, a gRNA incorporating two MS2 RNA aptamers at the tetraloop and stem-loop 2 and the MS2-P65-HSF1 plasmid which expresses the activation helper protein. We initially transduced 293 T cells with either the lenti MS2-P65-HSF1_Hygro or the lenti dCAS9-VP64_Blast plasmids together with the packaging plasmid psPAX2 and envelope plasmid pMD2.G to generate virus[28]. We then transduced the cancer cells with each lentivirus sequentially, selecting for cells with integration of virus using hygromycin and blasticidin. Cells stably resistant to both antibiotics were then used for transduction with the gRNA library.

**Generation of genome-wide CRISPR activation libraries and positive selection screens.** To generate the pooled lentiviral gRNA library of 70,297 guides targeting 18,965 coding genes, the lenti sgRNA(MS2)_puro pooled library was used to generate lentivirus from 293 T cells together with psPax2 and pMD2.G (Addgene #12260, #12259) packaging and envelope plasmids. The gRNA library lentivirus was then used to

transduce the cancer cell line stably expressing integrated for the lenti MS2-P65-HSF1_Hygro and the lenti dCAS9-VP64_Blast plasmids as described above. We transduced the cancer cells at MOI < 0.3 to ensure that most cells receive only one genetic perturbation. Cells were selected in puromycin for 10 days prior to the start of the drug resistance screen. The lenti MS2-P65-HSF1_Hygro, lenti CAS9-VP64_Blast, and lenti sgRNA(MS2)_puro plasmids were gifts from Feng Zhang (Addgene plasmid # 61426, #61425, #1000000074). Following selection, cells were maintained in culture for 14 days to allow for complete depletion of protein products of targeted genes, following which cells were treated with either DMSO (control), gefitinib or osimertinib (100 nM, drug replaced twice per week). After 21 days of drug selection, cells from each of the duplicate DMSO and treatment arms were harvested and submitted separately for PCR and Illumina sequencing.

**Illumina sequencing of gRNAs and statistical analysis.** Genomic DNA extraction and Illumina sequencing of gRNAs were conducted[29]. In brief, 72ug of total extracted DNA was used to set up 36 PCR reactions (2ug each) using 10uM concentrations of forward and reverse primers following which PCR products were purified using spin columns before a second PCR reaction was carried out to incorporate indexing primers for each sample. DNA was purified using SPRI beads and submitted for Illumina sequencing. Enrichment and depletion of guides and genes were analysed using the MAGeCK RRA (version 0.5.8) statistical package by comparing read counts from each cell line with counts from matching DMSO cells, after comparing each to counts from the plasmid gRNA library (https://sourceforge.net/projects/mageck/)[30]. MAGeCK RRA ranks sgRNAs based on their *p* values calculated from the negative-binomial model and uses a modified RRA algorithm named α-RRA to identify positively or negatively selected genes.

**Arrayed CRISPR screen**
**Preparation of Individual and Library of gRNA.** Synthetic two-part gRNA (cr:tracrRNA) supplied by Horizon Discovery (Cambridge, UK) was used for all experiments and prepared according to manufacturer instructions. In brief, individual tubes of crRNA were resuspended to 10 µM in the presence of an equimolar concentration of tracrRNA (Edit-R Synthetic tracrRNA, Horizon Discovery, Cambridge UK) in 10 mM Tris-HCl pH7.4. A bespoke arrayed library of 0.1 nmol crRNA for 155 genes (up to 4 crRNA designs per gene pooled per well) was supplied pre-dispensed into a 384-well acoustic plate (Echo PP-0200, Labcyte, San Jose, CA) and resuspended to an equimolar concentration of cr:tracrRNA of 2.5 µM by the addition of 40 µL duplex buffer (2.5 µM tracrRNA, 10 mM Tris.HCl pH7.4). Resuspended, duplexed cr:tracrRNA was used immediately following a 30 minute room temperature (RT) incubation or frozen at −80C for future use.

**Reverse Transfection of Synthetic gRNA.** On the day of screening, frozen cr:tracrRNA duplexes were thawed and allowed to equilibrate to RT prior to use. An acoustic dispenser (Labcyte Echo 555, San Jose, CA) was used to add 400 nl of duplexed library or 100 nl of control duplexed cr:tracrRNA into the wells of 384w plates (Cell Carrier Ultra, Perkin Elmer, Waltham, MA). Each plate included in the screen contained non-targeting controls (NTC) gRNA, gRNA for positive control genes (NF2, PTEN), and gRNA for biomarker controls (YAP1/WWTR1). Transfection solution (10 µL serum-free RPMI-1640, 0.75% [v/v] RNAiMAX, Thermo-Fisher, Waltham, MA) was added into each well (Multidrop Combi, Thermo-Fisher, Waltham, MA) and incubated at RT for 20 minutes. Adherent Cas9 expressing PC-9, HCC4006, and HCC827 cell lines were removed from the culture vessel with TrypLE, cell number counted, and 40 µL cell suspension was dispensed into assay plates. All cell types were plated at a density of 700 cells/well and incubated for 72 hr at 37 °C, 5% CO2.

**Treatment with EGFR inhibitor.** After 72 hours of culture, reverse transfected cells were treated with EGFR inhibitor. A Tecan D300e

Dispenser (Tecan AG, Switzerland) was used to directly add either 50 nM osimertinib or vehicle control (0.15% DMSO (Merck KGaA, Darmstadt, Germany) to the appropriate wells of the 384w plates. Plates were incubated for a further 24 hr at 37 °C, 5% CO2.

### Immunofluorescence staining, high-content imaging, and analysis.
After a total of 96 hr (72 hr reverse transfection followed by 24 hr EGFR inhibitor treatment), cells were fixed with 4% (w/v) PFA followed by a 2 hr RT incubation in blocking/permeabilization buffer (0.2% [v/v] Triton-X100, 2% [w/v] bovine serum albumin, PBS). Replicate groups of arrayed CRISPR edited plates from each cell type were stained either with a primary antibody solution containing biomarkers for components in the MAPK/ PI3K pathways (phospo-p44/42 (mAb E10 #9106), phospho-AKT (RmAb, D9E #4075, AF647 conjugated) and phospho-S6 (RmAb, D57.2 #9865, AF594 conjugated)) or HIPPO pathway (anti-YAP1/ WWTR1 (mAb D24E4, #8418). All primary antibodies were supplied from Cell Signaling Technology (Danvers, MA). Cells were incubated with primary antibody solutions at 4 oC overnight before washing 3x in blocking/permeabilization buffer (0.2% [v/v] Triton-X100, 2% [w/v] bovine serum albumin, PBS) followed by a 2 hr RT incubation with secondary staining solution made up in block/permeabilization buffer supplemented with 2 µg/ml Hoechst 33342 (Thermo-Fisher) and 4 µg/ml AlexaFluor 488 anti-mouse or 4 µg/ml AlexaFluor 647 anti-rabbit (both Thermo-Fisher, Waltham, MA). Cells were then washed 3x in PBS and sealed with black plate seals prior to imaging.

Images were acquired using a Cell Voyager 8000 (CV8000, Yokogawa Inc., Tokyo, Japan) with a 20× water objective and a 2×2 image binning; for each 384-well plate, a total of 4 field-of-view were captured at a single z-plane. Columbus (PerkinElmer, Waltham, MA) image analysis software was used to quantify cell number and mean fluorescence intensity per cell from the nuclear and cytoplasmic regions. The nuclear and cytoplasmic intensities were used to calculate the Nuclear: Cytoplasmic ratio (Nuc:Cyt) of YAP1/WWTR1 staining per cell. Data was imported into Genedata Screener v16 (Genedata AG, Basal, Switzerland) for normalization and quality control. Data for the MAPK/ PI3K biomarker set was expressed using a 2-point normalisation methodology where the NTC gRNA without inhibitor treatment represented a maximum signal (0) central reference and the NTC gRNA with inhibitor (50 nM osimertinib) condition was used as the minimum signal (−100) scale reference. Data for YAP1/WWTR1 nuclear translocation was expressed as Nuc:Cyt fold change over the NTC condition. Nuc:Cyt fold change was calculated for both with and without EGFR inhibitor treatment.

### Generation of isogenic cell lines with gene knockout or over-expression.
To generate gene knockout isogenic cell lines, stable Cas9-expressing cells were transduced with sgRNA targeting specific gene exons or non-targeting controls (NTC). Cells were transduced with the pKLV2-U6gRNA5(BbsI)-PGKpuro2ABFP-W vector (#67974, Addgene) or pKLV2-U6gRNA5(BbsI)-PGKpuro2AZsG-W (Addgene #67975) expressing gene-specific guide RNAs (Supplementary Data 17). Puromycin was used to select for transduced cells. Successful transductions were monitored using a MACS Quant Flow Cytometer (Miltenyi Biotec, Germany) and data was analysed in FlowJo. Lost expression of all target genes was confirmed by western blot 14 days post transduction (Supplementary Fig. 3).

Activation of MET gene expression from endogenous loci was achieved by using cells stably expressing MS2-p65-HSF1 and dCas9-VP64 as described earlier. Cells were transduced with commercially available vector LV06 (Merck, KGaA, Germany) expressing promotor-specific gRNAs (Supplementary Data 17). Puromycin was used to select transduced cells. Activation of expression of target genes was confirmed by western blot 14 days post transduction.

Activation of YAP1 and WWTR1 gene expression was achieved by synthesising the coding sequences of YAP1 (NM_001130145.3), WWTR1 (NM_015472.6) and cloning into the modified expression vector pKLV2-

EF1a-Cas9Bsd-W (#68343, Addgene, cloning and synthesis performed by GeneScript Biotech, Netherlands). The Cas9-Blasticidin expression cassette was exchanged for a YAP1 or WWTR1 coding sequence fused to a bleomycin resistance cassette. Vectors were transduced lentivirally as described earlier in this manuscript. Cell lines were selected using zeocin (500 µg/ml, Invitrogen, US). For inducible expression of YAP1 or WWTR1 WT or constitutively active mutant variants (5SA or 4SA, respectively) coding sequences were cloned into the *Age*I and *EcoR*I site of pTRIPZ (Horizon Discovery Group, UK). Vectors were transduced lentivirally as described earlier in this manuscript in PC-9 and HCC827 cells. Cell lines were selected using puromycin (1 µg/ml). Expression was induced by treating cells with indicated concentrations of doxycycline for the indicated time.

### Long-term clonogenic proliferation assays.
To measure long-term proliferation under different conditions 5000 cells were seeded into 6 well plates and 24 h later treated as indicated. Following the indicated treatments, media was discarded and cells were washed with PBS. Cells were fixed with 1 ml BD Cytofix (BD Biosciences, #554655) for 20 min at ambient temperature. Fixative was discarded and cells were stained with 1 ml 0.01% (w/v) crystal violet in dH2O for 30 minutes immediately. Then plates were washed with H₂O and air dried for 24 h. Plate scans were acquired using a GelCount (Oxford Optronix, UK) colony counter. Staining of wells was quantified by generating binary images and measuring area coverage using ImageJ software. At least two biological replicates were carried out consisting of at least two technical duplicates.

### Long-term live cell imaging.
To track cell proliferation over longer periods, cells were cultured in an Incucyte S3 microscope (Sartorius, Germany) for the indicated time. Initially, cells were seeded at low density into 96-well plates and grown until entering exponential growth (usually 3-4 days). Thereon, cells were treated as indicated, and media and treatment were renewed every 3 days. In the case of experiments with inducible constructs, expression was induced 3 days prior first drug treatment to establish stable expression conditions. Eight-phase contrast images of every well were acquired every 4 h with 20x magnification. Confluency was analyzed and averaged using the Incucyte S3 software. Each experiment was performed in triplicates and mean confluency was plotted using GraphPad prism.

### Short-term viability assays.
To measure short-term effects on viability under different conditions 2500 cells per well (PC-9, HCC827, HCC4006) were seeded into 96 well plates. Cells were treated as indicated 24 h after seeding. For determining viability CellTiter-Glo (Promega) was added to wells as per manufacturer's instruction. Luminescence was detected as a measure of viability using a CLARIOstar plate reader (BMG labtech). Results were analysed using GraphPad system software and are visualized as fractions or percentages of control.

### PDX Organoid derivation.
For organoid derivation from PDX tumors, tumors were cut into ~2–3 mm size pieces. After washing with PBS, the pellets were resuspended in 10 ml PBS containing 2.5 mg Liberase DH (Sigma, 7891) and rotated at 37 C for 30 min to 1 h to dissociate. The samples were collected by centrifugation and kept on ice for 7 min in 10 ml Ack Lysing buffer (Gibco, A10492-01). Then samples were washed with 3 ml PBS, filtered through 70 um Cell Strainer (Falcon 352350), and stored frozen in Recovery™ Cell Culture Freezing Medium (Thermo-Fisher, 12648010). Ex-vivo culture was carried out in Advanced DMEM/ F-12 medium[31].

For qPCR, organoids were treated for 48 h with DMSO, osimertinib at 160 nM or 500 nM. For IHC, organoids were treated similarly, but for 5 days. Prior to FFPE block preparation, organoids on chamber slide (Nunc, 154534) were fixed with 4% paraformaldehyde for 2 h at room temperature, washed with PBS, and transferred into a Cryomold (Tissue-tek) for embedding in Histogel (Thermo Scientific, HG-4000-012).

**TEAD reporter assay.** Cas9 expressing PC-9 and HCC827 cells were transduced with pLVdCIN containing a luciferase reading frame under transcriptional control of an 8xGTIIC TEAD consensus binding site (Addgene #34615)[9]. Transduced cells were selected using G418 (ThermoFisher Sientific, #10131035, 0.5 µg/ml or 1 µg/ml for HCC817 or PC-9, respectively). KO of NTC or NF2 was achieved as described earlier in this manuscript. To measure TEAD reporter activity in a multiplexed assay 2500 cells per well were seeded in 96 well plates. First, 72 h after seeding cells were stained using CellTiter-Fluor (Promega) according to manufactures instructions, and green fluorescence was detected as a measure of cell number using a CLARIOstar plate reader (BMG labtech). Second, cells were lysed using ONE-Glo EX Luciferase assay system (Promega) according to manufactures instructions and luciferase activity was measured using a CLARIOstar plate reader (BMG labtech). TEAD reporter activity was calculated by normalising luciferase activity to cell number measurement from the least technical duplicates. Shown are representative results from three independent experiments.

**Flow cytometry-based competitive proliferation assays.** For generation of YAP1 or WWTR1 KO and YAP1/WWTR1 double knockout cells sgRNAs targeting YAP1 or WWTR1 were cloned into the dual sgRNA expression construct pKLV2.2-h7SKgRNA5-hU6gRNA5-PGKpuroBFP-W (Addgene #72666). PC-9 NF2 KO cells were generated as described earlier, co-expressing ZsGreen as fluorescent marker. These cells were transduced with dual expression vectors containing sgNTC, sgYAP1, sgWWTR1 or a combination of both. Cells transduced with both vectors were co-expressing BFP and GFP as monitored using a MACS Quant Flow Cytometer (Miltenyi Biotec, Germany). Three days after transduction of dual guide vectors cells were co-cultured with control cells at defined ratios as monitored by flow cytomerty and treated with osimertinib or DMSO. Changes in ratio between isogenic KO populations and control populations as a measure of proliferation were monitored via flow cytometry at indicated time points and normalized to d0 ratios (Supplementary Fig. 11). Flow cytometry data was analysed using Flow Jo software. Experiments were performed as biological replicates consisting of three technical replicates.

**RNA-Sequencing and analysis.** RNA was extracted using the RNeasy 96 QIAcube HT Kit (Qiagen) on a QIAcube HT instrument and eluted in RNase-free water. RNA concentration was determined by Qubit (Invitrogen), purity was determined by NanoDrop 8000 (Thermo Scientific), and RNA integrity was measured using a 2100 Bioanalyzer (Agilent). All samples had a RIN of >7.0. Libraries were prepared using mRNA Stranded library preparation kit (Illumina) and subsequently quantified by Qubit and KAPA library quantification kit, ROX low (Roche). Library sizes were also determined by bioanalyzer. Paired-end sequencing was performed on a NovaSeq 6000 instrument (Illumina). The STAR software (v2.6.1d) was used to align reads to genome build hg38 (https://github.com/alexdobin/STAR/releases).

Data was analyzed by ROSALIND® (https://rosalind.bio/), with a HyperScale architecture developed by ROSALIND, Inc. (San Diego, CA). Reads were trimmed using cutadapt1. Individual sample reads were quantified using HTseq and normalized via Relative Log Expression (RLE) using DESeq2 R library. Read Distribution percentages, violin plots, identity heatmaps, and sample MDS plots were generated as part of the QC step using RSeQC6. DEseq2 was also used to calculate fold changes and $p$ values and perform optional covariate correction. Hypergeometric distribution was used to analyze the enrichment of Hallmark pathways from MSigDB. Enrichment was calculated relative to a set of background genes relevant to the experiment. Additionally, gene set enrichment analysis (GSEA) (16199517) was performed to query gene expression data for a Hippo gene signature of commonly induced target genes caused by YAP1 or WWTR1 overexpression[10].

To investigate YAP1 or WWTR1 specific gene sets, for each cell line three subsets of deregulated genes were defined (YAP1 unique, WWTR1 unique, YAP1/WWTR1 common genes). These groups were used as input for significant pathway enrichment focussing on the 'MSiGDB Hallmark 2020' signature with the Enrichr tool (https://maayanlab.cloud/Enrichr/)[32]. For significantly enriched signatures (FDR < 0.2) a combined 'enrichment score' is also calculated that takes both signature size and significance into account. We also used Enrichr to infer transcription factor (TF) activity based on differential expression of downstream transcripts in NF2 KO, WWTR1 OE, and YAP1 OE isogenic models of PC-9, HCC827, and HCC4006 cells. We used TRANSFAC_and_JASPER_PWMs, a curated collection of profiles of transcription factor binding sites for 222 transcription factors, to identify significantly activated TFs for each gene perturbation model ($p < 0.01$) (Supplementary Data 18).

**Analysis of published single-cell RNA-seq datasets.** scRNA-seq count matrices were obtained from GEO (GSE150949[14], GSE149383[15]) and processed following the standard Seurat workflow. In summary, gene counts were normalised using total counts per cell, multiplied by a factor of 10000 and log-transformed using the NormalizeData function, and finally scaled using ScaleData.

The 2000 top highly variable genes were selected using FindVariableFeatures function (method = 'vst'), and used as input for PCA. The first 20 PCs were then used for clustering and to calculate the UMAP representation of the data. Overall, the representation of the data recapitulated the results in the original manuscripts.

Gene scores were then calculated using the AddModuleScore function, and cells were grouped according to sample metadata, namely the day of collection; for Oren et al, day 14 cells were also grouped according to experimental sorting based on the level of cycling. Wilcoxon rank sum test between the first and each of the following days was performed. We calculated enrichment for a published Hippo transcriptional signature at each timepoint of osimertinib treatment[10].

**Real-time PCR.** RNA samples were harvested using RNeasy Plus Mini Kit (Qiagen). The RNA concentrations were measured using Nanodrop (Thermo Scientific). 1ug RNA was used as a template to set up cDNA synthesis reactions in 40ul using High-Capacity cDNA Reverse Transcription Kit (Thermo Fisher). Q-PCR reactions were set up in 20ul using 1ul cDNA as a template. TaqMan gene expression master mix (Catalog # 4369016) and Taqman gene expression assays were used: Human GAPDH (Hs99999905_m1); CTGF (Hs00170014_m1); CYR61 (Hs00155479_m1); AMOTL2 (Hs01048101_m1). The reactions were run in QuantStudio 7 Flex machine (Applied Biosystems). The gene expression levels were normalized to GAPDH as a housekeeping gene.

**Immunofluorescent staining.** For immunofluorescent analysis of YAP1/WWTR1 expression cells were seeded at low density into 96-well Cell Carrier Ultra plates (Perkin Elmer, US) and treated as indicated. Media and treatment were replaced every 3-4 days. After treatment, media was removed and cells were fixed at room temperature for 30 min in 4% para-formaldehyde (buffered in PBS). Next, cells were blocked for 2 h at room temperature in PBS + 0.1% Triton-X-100 (Sigma-Aldrich, US) and 1.1% BSA (Sigma-Aldrich). Cells were stained with 1:100 primary YAP1/WWTR1 antibody (CST, #8418) in blocking solution at 4 C overnight. Subsequently, cells were washed four times with 200ul blocking solution. Secondary antibody and nuclear staining were performed using Alexa Fluor Plus647 labelled goat anti rabbit IgG (ThermoFisher, A32733) diluted 1:500 in blocking solution with Hoechst33342 (Invitrogen, H3570, 1:20000) for 2 hours at room temperature. We washed cells twice with 200ul blocking solution and PBS. Per well 41 images were acquired in an automated way using an Operetta CLS High-Content Analysis System (PerkinElmer, US). Acquisitions were performed using 20x magnification and following filters for excitation/emission (355-385 nm/430-500 nm; 615-645 nm/655-760 nm). Images were analysed using Columbus software (PerkinElmer, US). Briefly, nuclear and cytoplasmic regions were defined by using

Hoechst33342 and YAP1/WWTR1 intensities, respectively. Nuclear and cytoplasmic regions were used to measure YAP1/WWTR1 intensities. Cells at image boarders were excluded from analysis. Cells were ranked according to cell size and cells in the lowest and highest deciles were excluded from the analysis. The nuclear/cytoplasmic YAP1/WWTR1 ratio was calculated, normalized to respective control, and visualized as violin blot. All experiments have been performed at least as biological replicates.

**Patient-derived xenografts.** EGFR mutant lung cancer PDX models were provided by Crown Bioscience (LU5221, LU11870), Champions Oncology (CTG-2548, -2531, -2939, -2803, -2934), or from the Pasi Jänne lab at Dana-Farber Cancer Institute (DFCI-357, -306, -364). All animal studies were performed according to the UK Home Office and IACUC guidelines. Animal studies were conducted in accordance with UK Home Office legislation, the Animal Scientific Procedures Act 1986, and the AstraZeneca Global Bioethics policy or Institutional Animal Care and Use Committee guidelines. Experimental work is outlined in project license 70/8894, which has gone through the AstraZeneca Ethical Review Process. The models were implanted subcutaneously in immunodeficient NSG mice (Jackson Labs). Mice with tumors reaching 400–600 mm$^3$ were started on osimertinib treatment at 25 mg/kg QD via daily oral gavage. This is equivalent to the clinical dose of the drug. Tumor volume was monitored with a caliper and the volume was calculated using the following formula: V = $(\pi/6)*$length$*$width$*$height. We demonstrated tumour regression during the 28 days of daily treatment. At the end of this period, untreated and osimertinib-treated samples were collected from $n = 3$ mice each, and tissue formalin-fixed and sectioned. These samples were provided for YAP1 and TAZ IHC.

**Immunohistochemistry (IHC).** Formalin-fixed, paraffin-embedded xenograft biopsies were sectioned at 4 micrometres and stained for YAP1 and WWTR1 IHC on the Leica Bond RX autostainer (Leica Biosystems, DE). Slides underwent 20 minutes heat-induced epitope retrieval at ER1 for YAP1 and 30 minutes at ER2 for WWTR1. This was followed by an endogenous peroxidase block (Leica Biosystems, DE) as well as a second blocking step for both proteins; YAP1: PBS, 2.5% normal horse serum (NHS), and 1% normal goat serum (NGS) (both from Vector Laboratories, CA, USA) and WWTR1: DAKO serum-free protein block (Agilent, UK). Anti-YAP1 D8H1X antibody (#14074 CST, MA, USA) at 0.07 µg/ml in PBS, 2.5% NHS and 1% NGS (both from Vector Laboratories, CA, USA) and anti-WWTR1 E8E9G antibody (#83669, CST, MA, USA) at 0.4 µg/ml in Ventana antibody diluent with caesin (Roche Diagnostics, UK) were used. DAB detection and haematoxylin counterstain was performed using the Bond Polymer Refine Detection (Leica Biosystems, DE). Slides were then dehydrated, coverslipped, and scanned at 400x magnification on the Aperio AT2 slide scanner (Leica Biosystems, DE). Mouse xenograft tissue sections using cancer cell lines with high and low expression of YAP1 and WWTR1 were used as positive and negative controls to confirm specificity of staining (High YAP1 – SW620; Low YAP1 – NCI-H526; High WWTR1 – SK-OV-3; Low WWTR1 – NCI-H526) as well as cell lines where each gene was selectively knocked out using CRISPR/Cas9.

**IHC image analysis and scoring.** HALO image analysis software (Indica Labs, NM, USA) was used to quantify percentage YAP1 and WWTR1 staining in tumour cells using the multiplex IHC algorithm and tissue segmentation classifier. Strong (3+), moderate (2+), weak (1+), and negative (0) intensity scores were given to both the nucleus and cytoplasm of tumour cells, and a H-score out of 300 was calculated independently for both cellular compartments using $[(\%1 + cells) + (\%2 + cells * 2) + (\%3 + cells * 3)]$.

**Statistics and reproducibility.** The name of each statistical analysis for specific experiments is indicated in each figure legend. All replicates were biological were nature i.e. separate experiments rather than technical

replicates. Statistical $p$ values are shown to at least 2 decimal places and are typically represented as either $p < 0.01$ (**) or $p < 0.001$ (***). When plotting data, all graphs include error bars.

**Reporting summary**
Further information on research design is available in the Nature Portfolio Reporting Summary linked to this article.

## Data availability
All Supplementary Data tables are available in the Supplementary Data excel file. The read counts tables for all CRISPR datasets and also the complete MAGeCK gene summary files for each experiment are available to download: https://az.box.com/s/bgxjw7l7pjkbektqw59h84iiz1anpjij. Raw FASTQ files from RNA-sequencing analyses in the paper are available in ArrayExpress in the study 'EGFR mutant human non-small cell lung cancer cell lines PC9, HCC827, HCC4006 after NF2 knockout, YAP1 or WWTR1 overexpression or osimertinib treatment.' under accession E-MTAB-13831. Uncropped and unedited blot/gel images for all figures is presented in Supplementary Fig. 12. All other data are available from the corresponding author (or other sources, as applicable) on reasonable request.

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

## Acknowledgements

We thank Oona Delpuech and Sara Talbot for supporting next-generation sequencing studies. We thank the flow cytometry, light microscopy, genomics and research instrumentation core facilities at the Cancer Research UK Cambridge Institute.

## Author contributions

M.P.: Data curation, formal analysis, validation, investigation, visualization, methodology, writing–original draft, writing–review and editing. J.B.: Data curation, formal analysis, validation, investigation, visualization, methodology. D.B*., S.P., A.F., J.B**., S.A., H.T., N.G., L.T., D.O., J.P, D.B., S.T., R.M., J.Y., C.C., S.D., M.M., M.G.: Investigation, visualization, methodology. J.P., D.B., M.A., A.S., M.L.G., H.T., FGC***: Data curation, formal analysis. S.D., M.M., J.M., K.M., J.B.: Resources, methodology. U.M.: Conceptualization, formal analysis, supervision, funding acquisition, methodology, writing–original draft, writing–review and editing. * Deepa Bhavsar, ** Jessica Bateson, *** Functional Genomics Centre.

## Competing interests

J.P., D.B., A.F., R.M., N.G., S.A., M.G., S.T., M.L.G., D.O., C.C., S.D., H.T., M.M., K.M., D.B., J.M., J.B. and U.M. are employees of AstraZeneca. All other authors declare no competing interests.

## Additional information

## Functional Genomics Centre

**Ultan McDermott** ◉¹ ✉, **Daniel Barrell** ◉¹ **& Carlos Company**¹

A full list of members and their affiliations appears in the Supplementary Information.

