## [Peer Review File · Communications Biology]

Reviewers' comments:

Reviewer #1 (Remarks to the Author):

In this study, authors performed loss of function and gain of function screens to identify pathways affecting EGFR inhibitor resistance in EGFR mutant lung cancer. Authors identified YAP signaling as an important pathway promoting cell survival following EGFR inhibitor treatment. Authors showed that YAP signaling was acutely activated upon EGFR inhibitor treatment, and inhibition of YAP signaling suppressed cancer cell resistance. The study suggests that YAP inhibitors should be explored as combination partners of EGFR inhibitor for EGFR mutant lung cancer. Overall, this is a nice and comprehensive study. CRISPR screens are well done. Demonstration of the importance of YAP signaling in EGFR inhibitor resistance is convincing. The paper is well written. I only have a few small points for authors to consider.

1. Line 115. I am not sure FOSL1 should be considered as a YAP pathway gene. Does overexpression of FOSL1 increase the expression of classical YAP target genes?

2. The finding that YAP signaling is increased upon EGFR inhibitor treatment is important. Authors have relied on YAP/WWTR1 nuclear localization and YAP target gene expression as readout (Fig. 5A, 5B). It would be more convincing if authors can measure the expression of WWTR1 by western blot and qRT PCR. Unlike YAP, which is primarily regulated by nuclear translocation, WWTR1 is primarily regulated by protein degradation. If authors' hypothesis is correct, the protein level, but not mRNA level, of WWTR1 should increase upon EGFR inhibitor treatment.

3. Fig. 2. It is somewhat unexpected that many genes unrelated to YAP signaling (MED12, SETD1B, EP300, BAD, and BAX) influenced YAP/WWTR1 nuc:cyto ratio. It is not clear how reliable the nuclear translocation assay is. It would be more convincing if authors can perform WWTR1 western blot for a few examples.

4. The title of the last section is "Hippo signaling is activated acutely in patient-derived models following osimertinib treatment". I am not sure the section title is supported by the data. In the CTG-2548 model, nuclear expression of YAP is increased, but the expression of WWTR1 is decreased. There is no difference in the LU5221 model. This is simply not conclusive.

Reviewer #2 (Remarks to the Author):

The manuscript by Pfeifer et al. used genome-wide CRISPR screens to identify pathways that cause resistance to the EGFR inhibitors in lung cancer cell lines, with a subsequent secondary screen with high-content imaging to detect activity of specific signal pathways they identified. The authors then mainly focused on one of the findings on YAP/Hippo pathway to characterize their mechanistic contributions and its potential as a combinational therapy target. The study is generally well done, and the screen appeared to have identified relevant genes as intended. However, the role of YAP activation in drug resistance process in general has already been widely reported in lung cancer (ref. 1-6) as well as other types of cancer (ref. 7), which unfortunately diminished the novelty of the main aspect of this manuscript. Nonetheless, the study adds more insights into the mechanism of resistance and specific contribution of Hippo pathway in lung cancer treated with EGFR-TKI. There are additional specific concerns on some of the presented data as listed below.

Major concerns:

1. In Fig 2E, the authors showed that the ratio of YAP/TAZ in nucleus to cytoplasm was increased after the EGFR-TKI treatment. However, it is not justified if the overall expression level should not be accounted for as well. In addition, it would be helpful to evaluate the level of p-YAP to further support their claim.

2. The screen was performed under 2-week exposure which would include the contributions from both the early persisting processes and later resistance mechanisms. The authors describe YAP nuclear localization/activation is an acute response 24h after EGFR-TKI treatment. At this time point, it is likely cells have already entered into a drug tolerant persister stage, where cells become slow-cycling. As YAP nuclear localization/activation is known to be associated with cell proliferation, how would the authors reconcile this observation? Or are the authors claiming YAP activation is a feature during the early phase of drug tolerance?
3. In Fig 3b-c, it is unclear how the nuclear to cytoplasm ratio of YAP expression was measured, assuming that this was done outside of the secondary screen in Fig 2. Besides, to support their claim that NF2 exerts its function through regulating Hippo pathway, the authors should also show by Western blot expression level of YAP in nuclei and cytoplasm separately, p-YAP level, and other regulators of Hippo pathway such as MST1/2, LATS1/2 in control and NF2 KO cells under the conditions of DMSO or Osimertinib at 24 h and day 7 (as in Fig 3c).
4. In Fig 3E, it appears that under DMSO treatment, there is a combinational effect of knocking out YAP and WWTR1. This suggests that YAP/WWTR1 is essential (assuming NF2 KO effects are negated by the double knockout) for their survival regardless of EGFR-TKI treatment. The authors need to clarify this point. In addition, under Osimertinib treatment, knocking out YAP alone (or WWTR1 alone to some extent) seems sufficient to inhibit cell proliferation and as good as double knockout. The authors need to explain the relationship between YAP and WWTR1 in this scenario as well. Secondly, it is important to show if YAP/WWTR1 expression and/or activity was increased by the treatment. The authors should also show if WWTR1 expression level changes, perhaps as a compensation, after the knock down of YAP. Since the molecular weights of YAP and WWTR1 are different, the difference of YAP and WWTR1 expression pattern can be relatively easily addressed by Western blot.
5. In Fig 4, the authors nicely showed the enrichment of the regulatory network by Hippo pathway. However, this was only done under non-treatment condition. Under different cellular contexts (even if NF2 knockout may partially mimic the Hippo inactivation induced by the treatment), the regulatory network might well be very different. To better address the contribution of Hippo pathway in the process of drug resistance, the authors should have profiled these under Osimertinib treatment. The observed enrichment for EMT may simply reflect the genetic perturbation rather than resistance mechanism. The authors should clearly state the caveats of these analyses.

Minor concerns:

1. The current description is misleading in terms of the relationship between Hippo pathway and YAP activation, and further drug resistance. The authors should clearly explain that Hippo pathway negatively regulates YAP nuclear localization and activity, as well as NF2 KO or YAP/WWTR1 overexpression is mimicking inactivation of Hippo pathway. For example, line 350, "hippo signaling activated" is the opposite direction of YAP target genes activation following Osimertinib treatment.
2. It would be more intuitive for the readers if the authors can show representative images of multiplexed IF in Figs 2A, 3C, showing the increased nuclear localization of YAP after treatment (at 24h and 7days)
3. From Fig 2B-D, the statement "The majority of genes that increased nuclear YAP1/WWTR1 in the cell lines did not alter pAKT or pERK signal intensity"(line 158) doesn't seem consistent with the data presented. It seems that genes that increase YAP nuclear translocation are clearly enriched for p-ERK, pAKT and p-S6 activation. The authors can perform formal enrichment analysis if so choose.
4. Fig 3F, the ctrl sample should be blotted on the same membrane in contiguous lanes with PC9 NF2 KO samples.
5. Fig 4B-E, Fig5B were not mentioned in main text. The authors should either add the descriptions on the main text or remove the figures.
6. Fig 5A, the control for this plot should be Osimertinib day 0 instead of DMSO day 1. Or DMSO versus Osimertinib at the same day would be a proper comparison.

References:

1. Park H S, Lee D H, Kang D H, et al. Targeting YAP-p62 signaling axis suppresses the EGFR-TKI-resistant lung adenocarcinoma. *Cancer Medicine*, 2021, 10(4): 1405-1417.
2. Kurppa K J, Liu Y, To C, et al. Treatment-induced tumor dormancy through YAP-mediated transcriptional reprogramming of the apoptotic pathway. *Cancer Cell*, 2020, 37(1): 104-122. e12.
3. Lee TF, Tseng YC, Nguyen PA, et al. Enhanced YAP expression leads to EGFR TKI resistance in lung adenocarcinomas. *Scientific Reports*, 2018, 8(1): 271.
4. Lee JE, Park HS, Lee D, et al. Hippo pathway effector YAP inhibition restores the sensitivity of EGFR-TKI in lung adenocarcinoma having primary or acquired EGFR-TKI resistance. *Biochemical and Biophysical Research Communications*, 2016, 474(1): 154-160.
5. Chung C, Lee D, Kim JO, et al. Targeting the hippo effector YAP overcomes the de novo and acquired resistances to EGFR-TKI in lung adenocarcinoma. *European Respiratory Journal*, 2016. 48 suppl 60.
6. Haderk F, Fernández-Méndez C, Čech L, et al. A focal adhesion kinase-YAP signaling axis drives drug tolerant persister cells and residual disease in lung cancer. *BioRxiv*, 2021: 2021.10. 23.465573.
7. Nguyen C, Yi C. YAP/TAZ signaling and resistance to cancer therapy. *Trends in Cancer*, 2019, 5(5): 283-296.

Genome-wide CRISPR screens identify the YAP/TEAD axis as a driver of persister cells in EGFR mutant lung cancer.

The text in the revised manuscript ('Article File') and this response letter is marked in red and magenta, respectively. We also indicate all changes to the manuscript by line number.

Reviewer #1 (Remarks to the Author):

In this study, authors performed loss of function and gain of function screens to identify pathways affecting EGFR inhibitor resistance in EGFR mutant lung cancer. Authors identified YAP signaling as an important pathway promoting cell survival following EGFR inhibitor treatment. Authors showed that YAP signaling was acutely activated upon EGFR inhibitor treatment, and inhibition of YAP signaling suppressed cancer cell resistance. The study suggests that YAP inhibitors should be explored as combination partners of EGFR inhibitor for EGFR mutant lung cancer. Overall, this is a nice and comprehensive study. CRISPR screens are well done. Demonstration of the importance of YAP signaling in EGFR inhibitor resistance is convincing. The paper is well written. I only have a few small points for authors to consider.

We thank the reviewer for the critical reading of our manuscript and providing insightful comments. We believe that their comments and suggestions have greatly strengthened our revised manuscript.

We have made substantial efforts to address most of the reviewers' comments and we believe that the revised manuscript is substantially improved as a result.

1. Line 115. I am not sure FOSL1 should be consider as a YAP pathway gene. Does overexpression FOSL1 increase the expression of classical YAP target genes?

We based our consideration of the FOS family of transcription factors (FOS, FOSL1, FOSL2 and FOSB) as YAP pathway genes based on a number of papers demonstrating that AP-1 is the second most abundant binding motif after TEAD for YAP/TAZ, and FOS family members are key components of the AP-1 complex. This is not an important part of the paper and we have not directly validated this as such but we felt it was important to highlight why it might be such a recurrent resistance gene from our CRISPR screens.

(a) In Shao et al (**Cell**, <https://pubmed.ncbi.nlm.nih.gov/24954536/>), a 15,294 ORF expression library was used to identify genes that rescued KRAS silencing & survival in KRAS mutant cell line following silencing of KRAS. The strongest hit was YAP1 overexpression (17-fold viability increase). Other hits were also WWTR1 (7-fold) and FOS (3.7-fold). FOSL1 and FOSL2 expression also increased viability. There was no evidence that this effect was mediated by TEADs (no rescue of KRAS suppression using a constitutively active TEAD vector) and instead this was demonstrated to be through AP-1 family transcription factors; expression of YAP1 or KRAS activated an

AP-1 luciferase reporter driven by a consensus AP-1 binding element and in addition expression of FOS rescued cells upon suppression of KRAS.

- (b) In Pham et al (**Cancer Discovery**, <https://pubmed.ncbi.nlm.nih.gov/33208393/>), the authors carried out ATAC-seq after YAP1 knockdown - showing decreased chromatin accessibility at TEAD- and AP1-binding sites upon YAP1 depletion. In addition, decreased FOSL1 phosphorylation in global phospho-proteomics following YAP1 depletion was observed. They also confirmed that TEAD directly interacts with FOSL1 through coimmunoprecipitation.
- (c) In Zanconato et al (**Nat Cell Biol**, <https://www.nature.com/articles/ncb3216>). Genome-wide association between YAP/TAZ/TEAD and AP-1 at enhancers drives oncogenic growth. In the NF2-null breast cancer cell line (MDA-MB-231), 70% of the YAP/TAZ/TEAD-occupied enhancers also contained AP-1-binding motifs, making AP-1 the second most abundant motif after TEAD. Sequential ChIP seq analysis for YAP followed by JUN suggested that both TEAD and AP-1 can bind to the cis-regulatory elements bearing TEAD and AP-1 composite sites at the same time and can physically interact with each other. AP-1 synergizes with YAP to increase oncogenic growth in mammary cells via activating target genes that control S-phase entry and mitosis. AP-1 has elevated activity in skin tumorigenic induced by chemical carcinogenesis. YAP/TAZ-deficient mice failed to produce tumors when subjected to chemical carcinogenesis, underscoring the importance of YAP/TAZ in AP-1-mediated tumorigenesis.
- (d) In Atkins et al (**Curr Biol**, <https://doi.org/10.1016/j.cub.2016.06.035>). An Ectopic Network of Transcription Factors Regulated by Hippo Signaling Drives Growth and Invasion of a malignant tumor model. In the NF2-null breast cancer cell line (MDA-MB-231), 70% of the YAP/TAZ/TEAD-occupied enhancers also contained AP-1-binding motifs, making AP-1 the second most abundant motif after TEAD. Sequential ChIP seq analysis for YAP followed by JUN suggested that both TEAD and AP-1 can bind to the cis-regulatory elements bearing TEAD and AP-1 composite sites at the same time and can physically interact with each other. AP-1 synergizes with YAP transcriptional regulation of AP-1 factors is also evolutionarily conserved in Drosophila. Activating transcription factor 3 (Atf3) is a direct transcriptional target of Sd, which is significantly upregulated in Ras driven tumour formation. Tumour specific gene expression in Drosophila is tightly regulated by a few key transcription factors, and AP-1 forms one of the major regulatory nodes. Loss of AP-1 or STATs can break this regulatory network by reducing the expression of tumour signature genes.

2. The finding that YAP signaling is increased upon EGFR inhibitor treatment is important. Authors have relied on YAP/WWTR1 nuclear localization and YAP target gene expression as

readout (Fig. 5A, 5B). It would be more convincing if authors can measure the expression of WWTR1 by western blot and qRT PCR. Unlike YAP, which is primarily regulated by nuclear translocation, WWTR1 is primarily regulated by protein degradation. If authors' hypothesis is correct, the protein level, but not mRNA level, of WWTR1 should increase upon EGFR inhibitor treatment.

We measured expression of total WWTR1 by western blot in PC-9, HCC827 and HCC4006 cell lines +/- osimertinib (figure below; red arrows) and NTC vs NF2 KO – new **Supplementary Figure 7C** – we confirm increased total WWTR1 protein expression following osimertinib treatment. We detected no change in mRNA expression upon osimertinib treatment in PC-9 cells using RNA-seq (Supp Table 12). We have added text to the manuscript to reflect this (Lines 314-320).

3. Fig. 2. It is somewhat unexpected that many genes unrelated to YAP signaling (MED12, SETD1B, EP300, BAD, and BAX) influenced YAP/WWTR1 nuc:cyto ratio. It is not clear how reliable the nuclear translocation assay is. It would be more convincing if authors can perform WWTR1 western blot for a few examples.

The Cas9-expressing PC-9 cell line was transfected with pooled synthetic gRNA (Synthego) targeting either MED12 or EP300 - lysates collected 72 hrs post transfection were probed for

the proteins below. We saw good knockdown efficiency for both genes but no effect on either WWTR1 or YAP1 total expression levels or indeed on EGFR signalling. In Supp Table 8, where we measured by nuclear translocation the YAP1/WWTR1 nuc:cyt ratio following knockdown of a number of resistance genes, we observed a modest increase in the ratio across multiple replicates (table below). I agree with the reviewer that the IF assays for nuclear translocation have a low dynamic range and therefore any in depth studies of a specific gene must also include determination of canonical Hippo pathway transcriptional targets as well as where available a TEAD binding reporter vector. This triad of tests was benchmarked against NF2 knockout in the paper and demonstrated its ability to readout for activation of Hippo signaling – for any genes below I would suggest a similar in depth approach would also be required.

Gene KO	YAP/WWTR1 nuc:cyt ratio normalised
NTC DMSO	1
NF2	1.74
FRMD6	1.66
RALGAPB	1.45
TAOK2	1.43
CSK	1.39
PPM1F	1.31
SETD1B	1.28
MED12	1.28
EP300	1.25
CAB39	1.23
WWC1	1.22
LATS2	1.21
C16orf72	1.20
BAD	1.15

4. The title of the last section is “Hippo signaling is activated acutely in patient-derived models following osimertinib treatment”. I am not sure the section title is supported by the data. In the CTG-2548 model, nuclear expression of YAP is increased, but the expression of WWTR1 is decreased. There is no difference in the LU5221 model. This is simply not conclusive. I agree that this is not conclusive and am happy to change to “**Increased nuclear YAP1 in a subset of patient-derived models following osimertinib treatment**” (Lines 354-365).

Moreover, we do specifically state in the discussion that “Of note, in the two PDX models we tested **only one demonstrated** YAP1/WWTR1/TEAD-dependent transcription in persister cells, highlighting that other mechanisms of persistence in EGFR mutant lung cancer must exist and need to be defined,” (Lines 427-430). In the ‘Limitations of the study’ section at the end of the paper, we highlight the need for increased biopsy of patients on treatment to better define whether pathways such as Hippo are indeed important clinically or not.

Genome-wide CRISPR screens identify the YAP/TEAD axis as a driver of persister cells in EGFR mutant lung cancer

The text in the revised manuscript ('Article File') and this response letter is marked in red and magenta, respectively.

Reviewer #2 (Remarks to the Author):

The manuscript by Pfeifer et al. used genome-wide CRISPR screens to identify pathways that cause resistance to the EGFR inhibitors in lung cancer cell lines, with a subsequent secondary screen with high-content imaging to detect activity of specific signal pathways they identified. The authors then mainly focused on one of the findings on YAP/Hippo pathway to characterize their mechanistic contributions and its potential as a combinational therapy target. The study is generally well done, and the screen appeared to have identified relevant genes as intended. However, the role of YAP activation in drug resistance process in general has already been widely reported in lung cancer (ref. 1-6) as well as other types of cancer (ref. 7), which unfortunately diminished the novelty of the main aspect of this manuscript. Nonetheless, the study adds more insights into the mechanism of resistance and specific contribution of Hippo pathway in lung cancer treated with EGFR-TKI. There are additional specific concerns on some of the presented data as listed below.

We thank the reviewer for the critical reading of our manuscript and providing insightful comments. We believe that their comments and suggestions have greatly strengthened our revised manuscript. We have made substantial efforts to address most of the reviewers' comments and we believe that the revised manuscript is substantially improved as a result. We appreciate that since we began this study, the novelty aspect has been diminished by publications from other labs – we suffered a loss of key lab members at a critical time owing to the unfortunate combination of Brexit and Covid which set us back considerably – c'est la vie!

Major concerns:

1. In Fig 2E, the authors showed that the ratio of YAP/TAZ in nucleus to cytoplasm was increased after the EGFR-TKI treatment. However, it is not justified if the overall expression level should not be accounted for as well. In addition, it would be helpful to evaluate the level of p-YAP to further support their claim.

Thank you for this suggestion. We have re-analysed lysates from osimertinib-treated PC-9 cells for pYAP1 S127 and total YAP1 as suggested and confirm loss of pYAP1 S127 following osimertinib treatment for 24h (new Supplementary Figure 7C).

In addition, we re-analysed the YAP/TAZ immunofluorescence data for this experiment for **Total/overall expression**, and there is **no evidence that the increase in nuclear expression is simply the result of an increase in total expression**:

2. The screen was performed under 2-week exposure which would include the contributions from both the early persisting processes and later resistance mechanisms. The authors describe YAP nuclear localization/activation is an acute response 24h after EGFR-TKI treatment. At this time point, it is likely cells have already entered into a drug tolerant persister stage, where cells become slow-cycling. As YAP nuclear localization/activation is known to be associated with cell proliferation, how would the authors reconcile this observation? Or are the authors claiming YAP activation is a feature during the early phase of drug tolerance?

This is a great point and speaks to the importance of using different timepoints to dissect out early vs late effects post EGFR inhibitor treatment, and also combining with scRNA-seq to allow accurate cell cycle analysis in subpopulations of cells. Our data (and that from single cell studies outline below) do support a role for YAP activation during the early phase of drug tolerance, and increases over time as YAP activated persister cells are enriched for.

1. Recent scRNA-seq papers of this same PC-9 cell line following treatment with EGFR inhibitors and looking at early, medium and late timepoints has confirmed that (a) there is strong cell cycle (G1) arrest in these cells as early as day 1 post treatment (see figures from papers below):

Oren, Y., et al. (2021). Cycling cancer persister cells arise from lineages with distinct programs. Nature 596, 576-582.

Aissa, A.F., et al. (2021). Single-cell transcriptional changes associated with drug tolerance and response to combination therapies in cancer. *Nat Commun* 12, 1628

2. Although the dogma would be that YAP activation is associated with cell proliferation, when we overexpressed YAP1 using a dox-inducible vector (Figure 3G) in PC-9 cells there was no change in proliferation over a 21-day time course (DMSO samples; blue line versus green – no Dox vs DOX).
3. YAP activation in our study causes EMT cell state switch – we believe this is part of the persistence/resistance phenotype – we cannot say whether this also causes the G1 arrest and whether the two are connected and of course, it is also possible that other programs active in these cells cause the cell cycle arrest
4. At early timepoints, likely mix of persisters plus arrested cells plus dying – need single cell resolution for this – importantly, in the Oren and Aissa single studies show above,

in addition to G1 arrest post EGFR inhibitor treatment in PC-9 cells, we also observed from re-analysis of the scRNA-seq data a significant enrichment for cells with a TEAD signature from as early as Day 1 post treatment (Supp Figure 8):

3. In Fig 3b-c, it is unclear how the nuclear to cytoplasm ratio of YAP expression was measured, assuming that this was done outside of the secondary screen in Fig 2.

In Methods, the section 'Immunofluorescence Staining, High-Content Imaging and Analysis' contains a full description of how YAP1 expression and the nuclear:cytoplasmic ratio is calculated – it is the same as per the secondary screen and I have added text in the figure legend to clarify.

Besides, to support their claim that NF2 exerts its function through regulating Hippo pathway, the authors should also show by Western blot expression level of YAP in nuclei and cytoplasm separately, p-YAP level, and other regulators of Hippo pathway such as MST1/2, LATS1/2 in control and NF2 KO cells under the conditions of DMSO or Osimertinib at 24 h and day 7 (as in Fig 3c).

Nuclear vs cytoplasmic protein fractions were not collected as part of this experiment as not considered critical to confirm NF2 loss as a driver of the Hippo pathway, primarily as loss of this gene has been shown to regulate Hippo over numerous publications and it is not a key part of the paper. We include additional new data below that NF2 KO increases expression of some canonical Hippo target genes as well as decreasing expression of pYAP S127 in the EGFR mutant cell line HCC827 – these have been added to **Supplementary Figure 5A**. Moreover, in Supp Figure 7A, NF2 KO was associated with increased expression of a Hippo gene expression signature as well as increased expression in Supp Figure 7E a luciferase TEAD reporter.

- NF2 KO increased YAP/TEAD downstream targets transcription

- NF2 is knocked out
- NF2 downstream marker (pYAP) is decreased

4. In Fig 3E, it appears that under DMSO treatment, there is a combinational effect of knocking out YAP and WWTR1. This suggests that YAP/WWTR1 is essential (assuming NF2 KO effects are negated by the double knockout) for their survival regardless of EGFR-TKI treatment. The authors need to clarify this point.

The reviewer is correct – in these DMSO-treated NF2 KO cells (left panel of Figure 3E), where we would expect activation of Hippo signalling and mediated through YAP/TAZ, there is indeed a dependency on YAP1 and WWTR1 as key mediators of this pathway – this would be expected and indeed is the clinical rationale for the recent clinical trials of TEAD inhibitors (Vivace Therapeutics) in NF2 mutant mesothelioma. We have added additional text in the manuscript for greater clarity (Lines 191-194).

In the right panel of this figure we show that even though NF2 KO causes strong resistance to osimertinib (we propose as a consequence at least partly of activating Hippo), this can be overcome by blocking the downstream activation of TEAD dependent targets through KO of either YAP1, WWTR1 or both.

In addition, under Osimertinib treatment, knocking out YAP alone (or WWTR1 alone to some extent) seems sufficient to inhibit cell proliferation and as good as double knockout. The authors need to explain the relationship between YAP and WWTR1 in this scenario as well.

Secondly, it is important to show if YAP/WWTR1 expression and/or activity was increased by the treatment. The authors should also show if WWTR1 expression level changes, perhaps as a compensation, after the knock down of YAP. Since the molecular weights of YAP and WWTR1 are different, the difference of YAP and WWTR1 expression pattern can be relatively easily addressed by Western blot.

We agree that the reciprocal relationship between YAP1 and WWTR1 is interesting. We did not see compensatory changes in total WWTR1 expression by Western Blot after the KO of YAP1 (**new Supp Figure 5B**) although interestingly in one of the PDX models following treatment with osimertinib, we observed that an increase in nuclear YAP1 expression was accompanied by a decrease in nuclear WWTR1 levels (Figure 6C).

5. In Fig 4, the authors nicely showed the enrichment of the regulatory network by Hippo pathway. However, this was only done under non-treatment condition. Under different cellular contexts (even if NF2 knockout may partially mimic the Hippo inactivation induced by the treatment), the regulatory network might well be very different. To better address the contribution of Hippo pathway in the process of drug resistance, the authors should have profiled these under Osimertinib treatment. The observed enrichment for EMT may simply reflect the genetic perturbation rather than resistance mechanism. The authors should clearly state the caveats of these analyses.

Thanks to the reviewer for drawing attention to the question of whether the observed transcriptional regulatory networks might be different in the presence of drug. Below is a graphical representation of the previous comparisons of RNA-seq data from the YAP1 or WWTR1 overexpression versions of the 3 cell lines. To address the reviewer's concern, we therefore analysed RNA-seq for each of the 3 cell lines for the YAP1 or WWTR1 overexpressing samples versus the EV and compared the regulatory networks **with and without Osimertinib as suggested**.

We observed that the **EMT gene expression** signature previously observed in all the YAP1 and WWTR1 OE (overexpression) isogenic models as well as post osimertinib treatment in Parental cell lines (in 2 of 3 cell lines) is **further enriched post treatment** in all of the cell lines and with both YAP1 and WWR1 – this has been added as a **new Supplementary Table 14b** in the paper. Additional text to reflect this has been added to the manuscript (Lines 277-280).

Minor concerns:

1. The current description is misleading in terms of the relationship between Hippo pathway and YAP activation, and further drug resistance. The authors should clearly explain that Hippo pathway negatively regulates YAP nuclear localization and activity, as well as NF2 KO or YAP/WWTR1 overexpression is mimicking inactivation of Hippo pathway. For example, line 350, “hippo signaling activated” is the opposite direction of YAP target genes activation following Osimertinib treatment.

We thank the reviewer for pointing out the potential issue with how we phrase the relationship between the Hippo pathway, YAP activation and activity of downstream transcription factors and programs. We have reviewed the manuscript and corrected these as suggested.

2. It would be more intuitive for the readers if the authors can show representative images of multiplexed IF in Figs 2A, 3C, showing the increased nuclear localization of YAP after treatment (at 24h and 7days)

We thank the reviewer for this suggestion. The IF (immunofluorescence) images below have been added to Supplementary Figure 2E and Figure 3C and the manuscript text and figure legends amended appropriately.

Figure 2E:

Figure 3C:

3. From Fig 2B-D, the statement “The majority of genes that increased nuclear YAP1/WWTR1 in the cell lines did not alter pAKT or pERK signal intensity” (line 158) doesn’t seem consistent with the data presented. It seems that genes that increase YAP nuclear translocation are clearly enriched for p-ERK, pAKT and p-S6 activation. The authors can perform formal enrichment analysis if so choose.

Thanks to the reviewer for drawing attention to this statement. We have performed a statistical 2-sided Fisher’s exact test comparing the gene KO’s that do and do not increase nuclear YAP1/WWTR1 expression for their effect on pERK, pAKT and pS6 expression. None of these p-values reached significance, defined here as $p < 0.01$. We have included a sentence to this effect in the manuscript (Lines 158-159).

4. Fig 3F, the ctrl sample should be blotted on the same membrane in contiguous lanes with PC9 NF2 KO samples.

We thank the reviewer for pointing out that the Control sample should be displayed on contiguous lanes – we have replaced Figure 3F with the complete WB below and changed to Supplementary Figure 5B:

5. Fig 4B-E, Fig 5B were not mentioned in main text. The authors should either add the descriptions on the main text or remove the figures.

The reviewer is absolutely correct in pointing out that we do not mention the these figures in the text. Apologies for the omission – we have addressed this in the relevant sections of the paper (Lines 257, 263-265, 314-317).

6. Fig 5A, the control for this plot should be Osimertinib day 0 instead of DMSO day 1. Or DMSO versus Osimertinib at the same day would be a proper comparison.

We thank the reviewer for his remarks. Under treatment conditions only a very small proportion of the cells (5-10%) survive and therefore we needed to seed initial cell numbers at high density. Therefore, after day 1 the DMSO treated wells become confluent and unsuitable to use as time-matched controls. Also, confluence is widely considered as modulating Hippo and cell signalling (<https://doi.org/10.1016/j.cell.2013.12.043>; <https://doi.org/10.1038/oncsis.2014.27>).

Therefore we used Day 1 DMSO treated, non-confluent cells as controls accepting that this is what is technically feasible under these experimental conditions. We have included a sentence to this effect in the paper to avoid any confusion and **amended Figure 5A** to reflect this.

References:

1. Park H S, Lee D H, Kang D H, et al. Targeting YAP-p62 signaling axis suppresses the EGFR-TKI-resistant lung adenocarcinoma. *Cancer Medicine*, 2021, 10(4): 1405-1417.
2. Kurppa K J, Liu Y, To C, et al. Treatment-induced tumor dormancy through YAP-mediated transcriptional reprogramming of the apoptotic pathway. *Cancer Cell*, 2020, 37(1): 104-122. e12.
3. Lee TF, Tseng YC, Nguyen PA, et al. Enhanced YAP expression leads to EGFR TKI resistance in lung adenocarcinomas. *Scientific Reports*, 2018, 8(1): 271.
4. Lee JE, Park HS, Lee D, et al. Hippo pathway effector YAP inhibition restores the sensitivity of EGFR-TKI in lung adenocarcinoma having primary or acquired EGFR-TKI resistance. *Biochemical and Biophysical Research Communications*, 2016, 474(1): 154-160.
5. Chung C, Lee D, Kim JO, et al. Targeting the hippo effector YAP overcomes the de novo and acquired resistances to EGFR-TKI in lung adenocarcinoma. *European Respiratory Journal*, 2016. 48 suppl 60.
6. Haderk F, Fernández-Méndez C, Čech L, et al. A focal adhesion kinase-YAP signaling axis drives drug tolerant persister cells and residual disease in lung cancer. *BioRxiv*, 2021: 2021.10. 23.465573.
7. Nguyen C, Yi C. YAP/TAZ signaling and resistance to cancer therapy. *Trends in Cancer*, 2019, 5(5): 283-296.

REVIEWERS' COMMENTS:

Reviewer #1 (Remarks to the Author):

Authors have addressed my concerns.

Reviewer #2 (Remarks to the Author):

The authors sufficiently addressed the concerns. Only suggestion is to remove the bottom panels for each cell line of Figure 2C/D as it is confusing and does not add any information than asterisks, or at least re-order the genes in the top panel to match with the bottom.